# Examining controls on peak annual streamflow and floods in the Fraser River Basin of British Columbia

Charles L. Curry[1,2] and Francis W. Zwiers[1]

[1]Pacific Climate Impacts Consortium, University of Victoria, Victoria, V8N 5L3, Canada
[2]School of Earth and Ocean Sciences, University of Victoria, Victoria, V8N 5L3, Canada

*Correspondence to*: Charles L. Curry (cc@uvic.ca). UNDER REVIEW in *Hydrological and Earth System Sciences*.
Version: Mar. 19, 2018.

**Abstract.** The Fraser River Basin (FRB) of British Columbia is one of the largest and most important watersheds in Western North America, and home to a rich diversity of biological species and economic assets that depend implicitly upon its extensive riverine habitats. The hydrology of the FRB is dominated by snow accumulation and melt processes, leading to a prominent annual peak streamflow invariably occurring in May-July. Nevertheless, while annual peak daily streamflow (APF) during the spring freshet in the FRB is historically well correlated with basin-averaged, 1 April snow water equivalent (SWE), there are numerous occurrences of anomalously large APF in below- or near-normal SWE years, some of which have resulted in damaging floods in the region. An imperfect understanding of which other climatic factors contribute to these anomalously large APFs hinders robust projections of their magnitude and frequency.

We employ the Variable Infiltration Capacity (VIC) process-based hydrological model driven by gridded observations to investigate the key controlling factors of anomalous APF events in the FRB and four of its subbasins that contribute nearly 70% of the annual flow at Fraser-Hope. The relative influence of a set of predictors characterizing the interannual variability of rainfall, snowfall, snowpack (characterized by the annual maximum value, $SWE_{max}$), soil moisture and temperature on simulated APF at Hope (the main outlet of the FRB) and at the subbasin outlets is examined within a regression framework. The influence of large-scale climate modes of variability (the Pacific Decadal Oscillation (PDO) and the El Niño-Southern Oscillation (ENSO)) on APF magnitude is also assessed, and placed in context with these more localized controls. The results indicate that next to $SWE_{max}$ (univariate Spearman correlation with APF of $\hat{\rho} = 0.64$ ; 0.70 (observations ; VIC simulation), the snowmelt rate ($\hat{\rho} = 0.43$ in VIC), the ENSO and PDO indices ($\hat{\rho} = -0.40$ ; $-0.41$) and ($\hat{\rho} = -0.35$ ; $-0.38$), respectively, and rate of warming subsequent to the date of $SWE_{max}$ ($\hat{\rho} = 0.26$ ; 0.43), are the most influential predictors of APF magnitude in the FRB and its subbasins. The identification of these controls on annual peak flows in the region may be of use in understanding seasonal predictions or future projected streamflow changes.

# 1 Introduction

## 1.1 Study domain and motivation

The response of nival watersheds across the Northern Hemisphere to ongoing warming of the climate system is a topic of intense interest to those interested in the water-climate nexus. Since the early reviews of Barnett et al. (2005) and Milly et al. (2005), which offered a global perspective, more recent studies have investigated the response on continental and regional scales. In mountainous regions that supply fresh water to downstream populated areas, reduced snowpack in recent decades has led to concerns about water availability for human consumption, agriculture and fisheries (Stewart, 2009; Jiménez Cisneros et al. 2014), while changes in the timing of the freshet and a widespread decrease in the snow-to-rain ratio (Danco et al., 2016) have implications for flood magnitude and frequency (Matti et al., 2016). An understanding of the mechanisms behind these changes has been greatly aided by the use of process-based hydrologic models (Park and Markus, 2014; Duan et al., 2017), which permit the analysis of a host of variables that respond to historical climate forcings via physically consistent relationships. We apply such a model to the Fraser River basin (FRB) of British Columbia (BC), Canada, a large, representative nival watershed, to study the dominant climatic controls on both observed and modelled hydrologic change.

The FRB is one of the largest watersheds draining the western slopes of the North American Cordillera, and home to both densely populated urban centres and a diversity of ecosystems closely linked to the economic prosperity of the region (Fraser Basin Council, 2010). The basin is also a focal point for salmon migration and spawning, constituting a key component of the highly valued commercial and subsistence fisheries of the region. The FRB lies between the Coast Mountains and the Continental Divide, with the headwaters of the Fraser in the northeast (~53°N, 118°W) and its outlet at the Pacific Ocean in the southwest (Fig. 1). Its vast 240,000 km$^2$ area encompasses a range of climatic zones, from the snowy mountains of the eastern Rockies to dry interior plateaus and wet fertile valleys nearest the Pacific west coast. The hydrological diversity of the basin is well described in Eaton and Moore (2010), who reviewed seasonal streamflow regimes at the catchment scale across BC, and also in Moore (1991), who focused specifically on the FRB. For the most part, streamflows are unregulated in the FRB, with the exception of Kenney Dam on the Nechako River in the northwest sector of the FRB (operational since the early 1950s) and several dams associated with the Bridge River power project in the Interior Plateau (completed in 1960). Regulated subbasins represent less than 10% of the total area of the FRB (Bawden, Burn and Prowse, 2015).

Streamflow in the interior catchments of the FRB is dominated by the snowmelt-fed spring freshet in April-July, leading to the usual characterization of the FRB as a nival basin. Hydrographs of major rivers in the FRB do not vary their form significantly from year to year due to the large amount of storage in the multi-year, high elevation snowpack. Indeed, the longest record of gauged flow at the southwestern outlet of the basin, at Fraser-Hope, has never recorded an annual peak

daily flow (APF) outside of the April-July period. Nevertheless, several catchments in the lower FRB, and to the west of the basin in the Coast Mountains and Western Cascades, exhibit annual streamflow peaks coinciding with the maximum of Pacific frontal rainstorm occurrence in October-December (Eaton and Moore, 2010; Padilla et al., 2015).

## 1.2 Previous studies of climate and streamflow change in the FRB

5        The focus of this work is upon the main drivers of the interannual (and to some extent, interdecadal) variability in annual peak daily flow (hereafter APF), and not upon secular trends in APF in the FRB and its subbasins. Nevertheless, as there is a growing body of work on such trends, which have implications for non-stationarity of the climate, we briefly review this research here, limiting the discussion to observations concerning APF magnitude. Significant long-term trends in temperature and precipitation have been detected over a network of meteorological stations in the FRB and its surroundings;

a summary of these trends is available in a recent report from the BC Ministry of the Environment (BC MOE, 2016) (in what follows, we average results for three regions used in the report which cover most of the FRB area: Sub-Boreal Interior, Central Interior, and Southern Interior). Annual (summer ; winter) mean temperatures have increased at the rate of +0.9 °C century$^{-1}$ (+0.7 ; +1.4 °C century$^{-1}$) over the 1900-2013 period, while minimum winter temperature has increased at the more rapid rate of 2.2 °C century$^{-1}$. The latter coincides with reduced winter snow depth ($-10\%$ decade$^{-1}$) and snow water

equivalent ($-6\%$ decade$^{-1}$; both over 1950-2014), mirroring a much larger-scale phenomenon occurring over Canada and the U.S. (Gan et al., 2013; Knowles, 2015) and much of the Northern Hemisphere (Rupp et al., 2013; Jeong et al., 2017). On the other hand, precipitation amounts have increased in the FRB over 1900-2013, by about +2% decade$^{-1}$ in spring, summer, and fall, with no significant change in winter precipitation (BC MOE, 2016). However, sparse station coverage in the early decades of the twentieth century casts doubt on the robustness of this result. Restricting attention to the period 1950-2014,

we used the regional data to compute precipitation trends of +3.3% decade$^{-1}$ in spring, +2.3% decade$^{-1}$ in fall and $-4.1\%$ decade$^{-1}$ in winter, with no significant trend detected in summer. Over the last century (1912-2012), an earlier onset of the half-annual streamflow volume date was detected in one subbasin of the FRB (Stellako, 9-day advance) as well as at the major outlet to the basin at Hope (6-day advance). The latter conclusion was reinforced by Kang et al. (2016), who noted a statistically significant advance, by ~10 days, of the APF at Fraser-Hope over the 1949-2006 period.

25        Recent decadal trends in the magnitude of annual and/or seasonal mean streamflow at gauge stations in BC and western Canada have been investigated by several authors (Pike et al., 2010; Bawden et al., 2015; DeBeer et al., 2016; BC MOE, 2016), with no significant trends detected at stations in the FRB. Hernández-Henríquez et al. (2017) conducted analyses on the FRB specifically, using a network of long-term measurements at 139 streamflow gauge stations available from the online Hydrometric Database (HYDAT; Water Survey of Canada, 2016). At Fraser-Hope specifically, Déry et al.

(2012) found an increasing trend in interannual streamflow variability between 1960-2005, in both annual and seasonal means, suggesting an increase in the frequency and/or intensity of low- and/or high-flow conditions, but with no detectable trend in annual mean streamflow at Hope.

With respect to trends in the APF specifically, Cunderlik and Ouarda (2009) found no significant trend at most Reference Hydrometric Basin Network (RHBN) stations in the FRB over the 1974-2003 period, but noted a weak trend (~1-5%) toward decreasing magnitude and earlier APF occurrence at two stations. A Canada-wide nonstationary analysis by Tan and Gan (2015), suggested that APF increased over recent decades at two stations in the FRB: the Stellako R. at Glenannan and Chilliwack R. at Chilliwack Lake. More aligned with our interests here, Burn and Hag Elnur (2002) found a weak negative correlation between APF and annual mean temperature at Quesnel in the FRB over the 1950-1997 period. We look for similar relationships between a range of climatic variables and APF at the scale of both the FRB and its major subbasins in Section 3 below.

Finally, it is important to recognize the influence of large-scale climate teleconnections on streamflow in the FRB. Previous researchers have investigated the influence of modes of large-scale climate variability on various river basins in western North America. Specifically, relationships between total and/or peak annual river discharge and the Pacific Decadal Oscillation (PDO), El Niño-Southern Oscillation (ENSO), and the Pacific North America index have been examined (Shabbar et al., 1997; Rood et al., 2005; Gobena and Gan, 2006; Bonsal and Shabbar, 2008; Whitfield et al., 2010; Gurrapu et al., 2016), while impacts of large-scale teleconnections on snowpacks in BC were examined by Moore and McKendry (1996) and Hsieh and Tang (2001). El Niño and La Niña typically occur every 3 to 5 years, often separated by 1 to 2 years of neutral conditions. The lower frequency PDO is a superposition of several interacting climate processes, including ENSO, mid-latitude ocean currents and atmospheric influences on mid-latitude sea surface temperatures (Alexander 2010). El Niño events are more likely to occur during the positive phase of the PDO, while La Niña events are more common during the negative phase.

During the negative (cold) phase of the PDO, and La Niña periods, winters in western Canada are typically cooler and wetter than average, with a larger snowpack at high elevations leading to higher annual discharge than average. Roughly opposite behaviour occurs during the positive (warm) PDO phase and occurrences of El Niño. Extending earlier work by Woo and Thorne (2003) and Thorne and Woo (2011), Gurrapu et al. (2016) determined that the PDO index is significantly anti-correlated ($p < 0.05$) with APF at eight hydrometeorological stations in the FRB, while APF is correlated with the Southern Oscillation Index (SOI), which tracks ENSO phase, at 11 stations (note that SOI is anti-correlated with the NINO3.4 index, and so also the PDO index). Specifically, both Thorne and Woo (2011) and Gurrapu et al. (2016) found that the observed APF was generally higher during the negative PDO phase and during La Niña years. For this reason, decadal trends in climatic variables, including streamflow, are sensitive to the phase of the PDO, which argues for a cautious interpretation of the results cited above.

The paper is structured as follows. Data sources, the VIC model and methods of analysis are described in Section 2. The main results of the study are gathered in Section 3, which begins with the regression analysis of observations before presenting insights from the VIC simulation results. Section 4 presents a few case studies that illuminate the precursors of

high streamflow in particular years, and also reinforce the regression-based results. We conclude in Section 5 with a short discussion of outstanding issues and conclusions.

## 2 Data and Methods

### 2.1 Study domain and observational data

The location of the FRB within Western Canada and BC is shown in Fig. 1. The observational data set used for analysis and for driving the hydrological model (see Sec. 2.2 below) is taken from the gridded data set of surface temperature and precipitation at daily temporal and 1/16° spatial resolution described by Schnorbus et al. (2011), hereafter referred to as PCIC-OBS. The original station data, which span the period from January 1950 to December 2006, are interpolated to the grid and corrected for elevation using the Climate Data for Western North America (Climate WNA) package (https://sites.ualberta.ca/~ahamann/data/climatewna.html; Hamann et al. 2013), based on very high resolution climatologies for the region that are developed with the Parameter-elevation Regressions on Independent Slopes Model (PRISM; Daly et al., 2008). Precipitation is not partitioned into rain and snow separately in PCIC-OBS. Since we are interested in investigating the relationship between rainfall and APF specifically, we need to estimate the rainfall-only portion, $R$. We do this using the empirical fit of Dai (2008), which relates the fractional rain frequency to surface temperature via a hyperbolic tangent function having four fitted parameters. We chose the parameter values corresponding to land-only, annual mean precipitation (seasonal fitted coefficients were also given by Dai (2008), but do not differ much from the annual values).

Daily streamflow data were obtained from the Water Survey of Canada (WSC) Hydrometric Database (HYDAT; Water Survey of Canada, 2016) for five hydrometric stations located within the FRB, as summarized in Table 1. The main outlet at Hope, which integrates the flows from all upstream locations, receives the most attention in the paper but four subbasin outlets are also considered. Three of these were selected on the basis of their leading contributions to the observed mean annual discharge at Fraser-Hope: Upper Fraser (29%), Thompson-Nicola (28%), and Quesnel (9%) (Kang et al., 2016). These subbasins are located in the eastern FRB, cover 45% of the total area, and represent nival environments. The smaller Chilko basin, by contrast, lies on the southwestern edge of the FRB and intercepts a significant amount of rain falling on the east-facing side of the Coast Mountains, making it a hybrid (nival-pluvial) catchment. Hence, this subbasin was included in an attempt to probe the sensitivity of streamflow to rainfall. Manual snow survey (MSS) SWE measurements taken at the beginning of each non-summer month (eight times per year) were obtained from the BC Snow Survey Network Program, distributed by the BC River Forecast Centre (http://bcrfc.env.gov.bc.ca/data/). The data do not permit the exact determination of the annual maximum SWE, so we extracted the 1 April SWE from each year for analysis. Data from 19 locations that are at least 81% complete spanning the period 1956-2014 were averaged to obtain an annual 1 April SWE time series for the entire FRB. The MSS stations are mainly located at high elevation sites in the Coast and Rocky Mountain ranges; exact locations are shown in Fig. 1 of Najafi et al. (2017). Another source of snow cover data, from automated snow

pillows at high elevation sites, was also examined. However, as only a few of these sites have records longer than 20 years (within the 1956-2006 period of PCIC-OBS), we decided to exclude them from the analysis.

We use the PDO and NINO3.4 indices to characterize the relevant large-scale climate modes affecting the region. The PDO index is derived as the leading principal component of monthly sea surface temperature (SST) anomalies in the North Pacific Ocean northward of 20 °N (http://research.jisao.washington.edu/pdo/PDO.latest.txt). For the corresponding predictor variable, we use the mean PDO index from the preceding November to March, following Gurrapu et al. (2016). The NINO3.4 index is calculated from the Hadley Centre SST and sea ice gridded data set, HadISST1, as the area average of SST from 5 °S-5 °N and 170-120 °W, available from http://www.esrl.noaa.gov/psd/gcos_wgsp/Timeseries/Nino34/. We use the mean value from June to November of the preceding year as a predictor. Note that this differs slightly from Gurrapu et al. (2016), who instead made use of the SOI over the same months (NINO3.4 and the SOI are strongly negatively correlated). A list of all variables analyzed may be found in Table 2.

## 2.2 Hydrological model and boundary forcing

The gridded temperature and precipitation data described above were used to drive the Variable Infiltration Capacity (VIC) hydrological model (Liang et al., 1994) over a large portion of BC, including the FRB, from 1950-2006. The VIC model is applied at a horizontal resolution of 1/16° (~5-6 km, depending on latitude), and solves the one-dimensional water and energy balance equations within each grid cell at a daily time step (with the exception of the snow sub-model, which runs at hourly resolution). The VIC implementation used in this study incorporates five elevation bands corresponding to 200 m vertical resolution, with the number of elevation bands in any one 1/16° grid cell depending on the topography within that cell. Each VIC grid cell can be assigned up to eight major vegetation classes, with a fractional cell area assigned to each, and these land cover fractions are identical for each elevation band within a VIC grid cell. Three of the input PCIC-OBS fields, maximum and minimum temperature and precipitation, are vertically interpolated to provide values for each elevation band (the other input field, surface wind speed, is not interpolated). In VIC, precipitation type (rain or snow) at the grid scale is determined by fixed temperature thresholds, with snow falling when $T < 0$ °C, rain when $T > 6$ °C, and a linearly interpolated mix of the two at intermediate temperatures. While this differs somewhat from the partitioning applied to the PCIC-OBS precipitation (Sec. 2.1) used in the regression analysis of Section 3 for both observations and models, the treatments are sufficiently similar that we do not expect a significant bias to arise. Snowpack is represented by a two-layer scheme—a thin surface overlying a thick deeper layer—subject to mass and energy balance, similar to other cold land process models. The snow albedo parameterization in VIC is based entirely on snow age, with a different albedo decay rate used during accumulation and snowmelt seasons. Interested readers are directed to Andreadis et al. (2009) for further details.

Glaciers are not explicitly parameterized in VIC; however, the version of VIC used here produces a perennial, accumulating snowpack at several high-elevation grid cells, many of which coincide with the locations of glaciers in the real system (also noted by Islam and Déry, 2017, who used a version of VIC with lower horizontal resolution but finer vertical resolution). These anomalous "glacier" cells cause the annual maximum of the basin average SWE, SWE$_{max}$, to increase

approximately linearly with time $t$: i.e., $SWE_{max} = at + SWE_0$, where $SWE_0$ is the value of SWE at an affected cell at the start of the simulation ($t = 0$) and $a$ is a fitted trend. The linear increase in $SWE_{max}$ is clearly unrealistic and is not seen, e.g., in the corresponding observed 1 April SWE time series averaged over the FRB, which has a trend indistinguishable from zero. In order to reproduce this basin-averaged trend of approximately zero in the VIC model, we masked out the affected

cells according to the criterion $a > 60$ mm y$^{-1}$, with the slope $a$ determined from a linear least squares fit over the 1950-2006 period of the calibrated VIC simulation (see below). In addition, we confirmed a posteriori that the removal of these anomalous cells had little effect on the correlation structure of the basin average $SWE_{max}$ with any of the variables examined in Sec. 3 below.

Soil parameters in VIC are defined for each grid cell. Soil classification and parameterization is based primarily on

physical data from the Soils Program in the Global Soil Data Products CD-ROM (GSD 2000), which originate from a global pedon database produced by the International Soil Reference and Information Center (ISRIC) (Batjes 1995) and the FAO-UNESCO Digital Soil Map of the World (DSMW) (FAO 1995). Although VIC contains a parameterization for seasonally frozen soils (Cherkauer and Lettenmaier 1999), it increases the computation time significantly and so was not activated (the simulation we analyzed was originally produced for a different purpose). Surface and subsurface runoff are generated for

each grid cell, and subsequently directed into a surface routing network (Lohmann et al., 1996, 1998; Schnorbus et al., 2011). Simulated streamflow can be extracted at grid cells representing outlets of the FRB or any one of its subbasins, which can be evaluated against stream gauge measurements.

VIC has been calibrated and evaluated in the FRB and its subbasins (Schnorbus et al., 2010; Shrestha et al., 2012; Shrestha et al., 2014) and also in other nearby hydrological basins (Schnorbus et al., 2011). Recently, Islam and Déry (2017)

used a lower horizontal resolution version of VIC (1/4°) to study its sensitivity to several different gridded input data sets, including PCIC-OBS. They found that while driving VIC with PCIC-OBS tended to overestimate SWE, the resultant hydrographs for the FRB and its basins were in better agreement with observations than when competing data products were used as driving data sets. VIC model output variables, including SWE, total column soil moisture (both over the period 1950-2006), and routed streamflow at the location of WSC stations (1955-2004), were obtained from the Pacific Climate

Impacts Consortium Data Portal (Pacific Climate Impacts Consortium, 2014).

### 2.3 Selection of predictors and analysis methods

We use regression models to study the relationship between APF at Fraser-Hope and climate variables, termed "predictors", that may influence APF. Predictors were selected based on physical intuition, inspection of the relevant literature and initial exploratory data analysis. Prior studies that were particularly helpful in this regard were Gurrapu et al.

(2016) who examined the influence of the PDO on streamflow in Western Canada, Jenicek et al. (2016) who examined the influence of snow accumulation and other variables on summer low flows, Coles et al. (2017) who studied snowmelt-runoff generation on Canadian prairie hillslopes, and Wever et al. (2017) who conducted model simulations of the joint effect of snowmelt and soil moisture on streamflow in a Swiss alpine catchment. The predictor variables chosen are listed in Table 2.

Both univariate and multivariate linear regression models were constructed to explore predictor-predictand relationships. The APF was determined as the maximum annual value of the running 3-day mean discharge at Fraser-Hope. Most of the predictors are represented as basin averages, but two, the ENSO and PDO indices, describe large-scale, multi-year climate modes of variability. The influence of large-scale climate can be manifested as nonstationary behaviour (e.g., trends) in the predictand and/or predictors. Since regression models are sensitive to both trends and autocorrelation in the underlying time series, all univariate regressions were checked for the presence of both (e.g. see Shumway and Stoffer, 2017, and specifically their online supplement, https://onlinecourses.science.psu.edu/stat510/node/53).

Specifically, for the predictand (APF) $Y_t$ and each predictor $X_t$, we first fit the models (by linear least-squares regression):

$$Y_t = \alpha_0 + \alpha_1 t + y_t , \qquad\qquad X_t = \beta_0 + \beta_1 t + x_t , \qquad\qquad (1)$$

where $y_t$ and $x_t$ are the corresponding detrended annual time series. If, as determined from the fit, there was no statistically significant temporal trend (with $p$-value $< 0.05$) in either $Y_t$ or $X_t$ , then the latter were used in the subsequent analysis in place of $y_t$ and $x_t$. Trends were detected in certain time series, as reviewed in Sec. 3.1 below. Regression models including variables with significant trends were generally found to have inflated correlation coefficients (e.g., Pearson's $R^2$ or Spearman's $\hat{\rho}$) compared to those using the same variables after detrending. The analysis proceeds using the fitted linear regression model to the detrended series $y_t$ and $x_t$, i.e., $\widetilde{y}_t = \gamma_0 + \gamma_1 x_t$ . If a statistically significant relationship was found (again with $p$-value $< 0.05$), then the associated correlation coefficient was taken to be a conservative measure of the relationship between the predictand and predictor. We say "conservative" because it is possible that the residuals $\varepsilon_t = y_t - \widetilde{y}_t$ may yet possess an autoregressive structure, i.e. $\varepsilon_t = \phi_1 \varepsilon_{t-1} + \phi_2 \varepsilon_{t-2} + ... \phi_i \varepsilon_{t-i} + w_t$ , where $\phi_1, \phi_1, ..., \phi_i$ are constants ($\phi_i \neq 0$) and $w_t$ represents white noise (Shumway and Stoffer, 2017). If $\varepsilon_t > 2/\sqrt{N_{yr}}$ at any lag $i > 1$, where $N_{yr}$ is the length of the time series, then further analysis would be required to fully specify the regression model. However, including autoregressive terms as predictors in the model will only increase the explained variance (e.g., Pearson's $R^2$), so we can be confident that the correlation coefficient after detrending, but without including autoregressive terms, underestimates the correlation coefficient. In practice, however, autocorrelation was detected only in a few cases (using the function `acf2` in the R module `astsa`), and principally amongst predictors themselves, not in the regressions of APF on predictors.

Ultimately, the nonparametric Spearman's rank correlation, with sample estimator $\hat{\rho}$, was used to characterize the interannual regression results as it makes no assumptions regarding the distribution of the climatic and hydrologic data. A correlation matrix of $\hat{\rho}$ was constructed by repeated univariate regression over all variables (including detrending where necessary), and the results summarized in the correlograms presented in Section 3. Another nonparametric measure, the Theil-Sen slope, is used to calculate temporal trends of the predictand and predictors in Sec. 3.1. The Theil-Sen slope is a median of slopes calculated for each pair of data points and is less sensitive to outliers than the standard least-squares regression line (Sen, 1968). Multiple linear regression (MLR) analyses were also conducted by including all possible combinations of the predictors listed in Table 2. We retained the MLR model that featured: 1) the most variables ($N$) with

partial *p*-values less than 0.05 ($N_{sig}$), and; 2) a Pearson-adjusted coefficient of determination $R_{adj}^2$ larger than any MLR with $N < N_{sig}$ predictors. We also tried adding predictors in a stepwise manner, obtaining similar results.

In order to discern differences between data samples drawn from potentially distinct distributions, we apply a permutation or resampling test at several points in the analysis. For two input data sets, $X$ and $Y$ of length $n_X$ and $n_Y$, respectively, the test statistic $S \equiv \mathrm{mean}(X)/\mathrm{mean}(Y)$ is calculated. The two sets are then combined, i.e., $Z \equiv X \cup Y$, and a large number (we chose $N = 10^4$) of paired, random samples of lengths $n_X$, $n_Y$ are drawn from $Z$. For each sample ($x_i$, $y_i$), the same statistic $S_i$ is recalculated, and the number of excedences, $N_{exc}(S_i > S,)$ summed over all $i = 1, \ldots, N$ samples. If the ratio, $p = N_{exc}/N$ is sufficiently near either 0 or 1, then the original test statistic $S$ is taken to indicate a significant difference between the samples at the $p$ or $1 - p$ level. We adopt the commonly used threshold value of $\min(p, 1 - p) = 0.05$.

Finally, while most predictors are well characterized by their annual maximum or seasonally averaged values, daily rainfall needs special attention due to its high temporal variance even when averaged over the entire FRB. To better identify significant, multi-day rainfall episodes, we computed the current rainfall index (CRI), after Fedora and Beschta (1989) and Smakhtin and Masse (2000):

$$\mathrm{CRI}_t = K \times \mathrm{CRI}_{t-1} + R_t = R_t + K\,R_{t-1} + K^2\,R_{t-2} + \ldots, K < 1 \tag{2}$$

The CRI incorporates both the most recent daily rainfall amount ($R_t$) and an exponential decay in the weight given to previous rainfall events. $K$ is the daily recession coefficient and reflects the storage capacity of the basin which depends somewhat on its physical properties (e.g., topography and area). We set $K = 0.9$ following a previous application in Pacific Northwest basins (Fedora and Beschta, 1989). The CRI proves useful for investigating intra-annual correlations between $R$ and daily streamflow, which is covered in Sec. 3.3.

## 3 Results

### 3.1 Trends in observed time series

We begin by searching for temporal trends in the input time series. These are of intrinsic interest, but also for interpreting the results of the univariate and multivariate regression models later in this section. In addition to APF, we computed linear trends using the Theil-Sen slope estimator for the eight observed variables listed in Table 2 over the common period of record 1956-2006, with the results summarized in Table 3. Of the nine variables, four display statistically significant trends at the $p < 0.05$ level: APF, freezing degree days (absolute value of the sum of negative daily mean $T < 0$ °C from 1 October − 31 March, hereafter FDD), April-June mean temperature ($T_{amj}$), and the PDO index. The correlation coefficient $R^2$ is small for all the fits ($R^2 = 0.10-0.16$), indicating that the trends are modest compared to the scatter in the annual data. The decreasing trend in FDD and increasing trend in April-June mean temperature (0.26 °C decade$^{-1}$) are qualitatively consistent with the regional temperature trends summarized in Sec. 1.1. The trend in APF of −37 m$^3$ s$^{-1}$ y$^{-1}$, or −4.3% decade$^{-1}$ is of the same sign but lower magnitude than that estimated for annual mean flow at Hope by BC MOE (2016) (−5.7% decade$^{-1}$ between 1958-2012). However, it is worth noting that over the entire 1912−2014 period of the gauge record at Fraser-Hope,

there is no significant trend detected in APF, even at the less conservative *p*-level of 0.1. The same is true of the PDO index over its much longer period of record (1900-2015). Nevertheless, we detrended the 1956-2006 observed time series of APF before computing the correlograms shown later in this section.

## 3.2 Influence of large-scale climate modes on observed streamflow at Fraser-Hope

Before investigating relationships between climate and APF at the basin scale, we examine the influence of the PDO and ENSO on the observed APF at the Fraser-Hope stream gauge station. This station was not included in the Gurrapu et al. (2016) study, but we derive results similar to theirs at other stations, as shown in Fig. 2. The figure shows that higher APF is associated with negative (cold) phases of the NINO3.4 (Fig. 2a) and PDO (Fig. 2b) indices over the 103-year record (1912-2014) at Fraser-Hope. While the statistical relationships between the large-scale climate modes and the APF are robust

(Spearman correlation of $\hat{\rho} = -0.40$ for NINO3.4, $\hat{\rho} = -0.35$ for PDO, both with $p < 10^{-3}$), Figs. 2a and b show that the largest APF in the record (1948; shown in boldface font on the plot) occurred during a neutral PDO phase and weak La Niña conditions, as did the fifth- and sixth-largest (1964 and 1997, also in bold). The second- to fourth-largest APFs (1972, 1950, 2012) occurred during strong negative PDO phases but weak La Niña conditions. This suggests that while APFs occurring during a negative PDO or La Niña phase may be larger than average, the very largest APFs may be influenced by more local

drivers, of either climatic or non-climatic nature (e.g., elevation and aspect). Indeed, our results indicate that the PDO and NINO3.4 indices explain only $\sim \hat{\rho}^2 \sim 0.12-0.16$ or 12−16% of the variance (with considerable interdependence between the two modes). Correlations of similar magnitude and sign were found at many more stations within the FRB by Thorne and Woo (2011).

Another way of visualizing streamflow sensitivity to ENSO and PDO phase is exhibited in the percentile plots of

Fig. 2c and d. These plots can reveal differences in sample distributions more clearly than a histogram or density plot. For example, for the PDO the time series of APF was first divided into years with positive ($PDO_{pos}$) and negative ($PDO_{neg}$) PDO index. Then a resampling (permutation) test was applied to check whether the test statistic $S = [mean(APF)]_{PDO,neg}/[mean(APF)]_{PDO,pos}$ differed from that calculated from $10^4$ random samples of the combined (both positive and negative PDO phase) data set (Sec. 2.3). Fig. 2d shows that the APF in years with negative (cold) PDO phase is

significantly larger than in years with positive (warm) PDO phase, consistent with the relationship seen in the scatter plot (Fig. 2b). Similar results were found for ENSO, as seen in Fig. 2c.

## 3.3 Basin-scale relationships amongst observed variables

To initiate our investigation of basin-scale drivers for APFs, we revisited the observed data. Specifically, we regressed the

observed APF at Fraser-Hope against FRB-wide averages of the observed predictors in Table 2. Fig. 3b shows a correlogram of the univariate regression results for all observed variables over the common period of 1956-2006. Variables with detected trends over this period (Sec. 3.1) were detrended before constructing the correlogram, as described in Sec. 2.3. We

summarize the results for individual predictors below, before looking at their joint influence on APF at the end of the subsection.

### 3.3.1 Snow water equivalent (SWE)

As expected given the mainly nival character of the FRB, we find that interannual variations in 1 April SWE and APF are
strongly correlated, with a Spearman correlation of $\hat{\rho} = 0.64$ over the common period of 1956-2014 of the MSS and WSC time series (Fig. 3a). This figure shows the raw data, while Fig. 3b shows the correlogram of all variables over the shorter common period of 1956-2006, including detrending where necessary. These differences lead to a somewhat smaller $\hat{\rho} = 0.51$ between 1 April SWE and APF. Nevertheless, the conclusion that years with higher than average SWE tend to produce higher peak streamflow at Fraser-Hope remains robust.

### 3.3.2 Temperature

Several diagnostics of the effect of surface air temperature $T$ on APF were examined. First, we computed FDD from the PCIC-OBS data averaged over the entire FRB. This helps determine whether exceptionally cold winters correspond to an unusually large snowpack, potentially producing an enhanced snowmelt contribution to the spring freshet. Second, we calculated the April-June mean temperature $T_{amj}$, which roughly coincides with the interval between the time of 1 April SWE
and APF. Finally, we computed the warming rate over the freshet period, $dT/dt$, by computing the linear least-squares slope of $T$ between 1 April and the date of APF, or annual peak date (APD). Over the 1956-2006 period at Fraser-Hope, the APD ranges from 16 May to 23 July, with a median of 13 June. The corresponding warming rates in different years range from 0.093 to 0.26 °C day$^{-1}$ (median = 0.16 °C day$^{-1}$), implying a typical warming over the median freshet (under the linear approximation) of ~ 11 °C.

The correlations of these variables with APF are displayed in the correlogram of Fig. 3b. Here it is seen that amongst these variables, April-June mean temperature is anti-correlated with APF ($\hat{\rho} = -0.42$), while FDD and $dT/dt$ are positively correlated with APF [$\hat{\rho} = 0.21$ (not significant) and $\hat{\rho} = 0.26$, respectively]. All else being equal, an unusually cold winter (high FDD) would be expected to result in a larger snowpack and a larger snowmelt contribution to streamflow consistent with the positive correlation found between FDD and APF. By the same reasoning, an unusually cold spring (low
$T_{amj}$) resulting in a delayed, but more rapid snowmelt during the freshet, would again be expected to increase that year's APF. In years with normal summer $T$, this situation would produce a larger than normal warming rate, consistent with the positive correlation found between $dT/dt$ and APF. Finally, we note from Fig. 3b that a positive correlation is seen between $T_{amj}$ and $dT/dt$ ($\hat{\rho} = 0.30$), implying that high spring temperatures are associated with rapid warming.

### 3.3.3 Rainfall

As mentioned in Sec. 2.3, daily rainfall $R$ shows a high temporal variance even when averaged over the entire FRB, which could introduce spurious noise into the regressions. For this reason, we looked at three integrated forms of $R$: 1) the summed rainfall over the winter months, $R_{Oct-Mar}$; 2) the sum of rainfall between 1 April and the day of APF, $R_{Spring}$; and 3) the sum of

$R$ over the 15 days prior to the APD, $R_{APF}$. These metrics offer differing probes of the influence of rainfall on the APF at different lags and temporal resolution. However, none of the rainfall measures displayed a significant relationship with observed APF (Fig. 3b), reinforcing the designation of the FRB as a primarily nival basin. Indeed, $R_{APF}$ was completely unrelated to APF ($\hat{\rho} = 0.02$, not shown in Fig. 3b). Although severe flooding in several small catchments in the U.S. Pacific Northwest has been attributed to rain-on-snow events (McCabe et al., 2007; Surfleet and Tollos, 2013), we see no evidence

of this in the much larger FRB.

Unlike SWE, the basin-averaged, annual maximum CRI, $CRI_{max}$, generally occurs during summer and fall (calendar days 160 to 301), although multiple peaks in a given year are often observed (see ff., Fig. 9). This contrasts with the narrower distribution of APF dates occurring in spring-summer (days 136-204), suggesting a weak relationship may exist between the two on the seasonal time scale.

Focusing exclusively on annual maxima or summed values of rainfall and APF might cause us to miss important relationships between rain and discharge on shorter time scales. To investigate these types of linkages, we interrogated the respective daily time series. Specifically, we computed the annual cross-correlation function between deseasonalized streamflow ($X$) and CRI ($Y$) anomalies for each year $i$:

$$\hat{\rho}_{XY,i}(\tau) = \frac{\text{mean}[X'_i(t)Y'_i(t+\tau)]}{\hat{\sigma}_{X,i}\hat{\sigma}_{Y,i}}$$

(3)

where $X'_i(t)$ is the anomaly time series of $X$ (i.e. difference between $X$ and its multi-year mean annual cycle) in year $i$, $\hat{\sigma}_{X,i}$ is the standard deviation of $X'_i$ (and similarly for $Y$), and $\tau$ is the applied lag (days) between $X'_i$ and $Y'_i$, ranging over $\pm183$ days. If $\max(\hat{\rho}_{XY,i}) > \delta$ at $\tau > 0$, where $\delta = 2/\sqrt{N} = 2/\sqrt{365}$, then CRI both leads and positively correlates with streamflow in that year, suggesting a causal relationship. The results are displayed in Fig. 4. Each point in the scatterplot represents

$\max(\hat{\rho}_{XY,i})$ calculated over $(365-\tau)$ pairs taken from the two daily time series for the $i^{th}$ calendar year. Fig. 4 shows that about half (26 of 50) of the years exhibit a significant relationship at some time during the year, with $\max(\hat{\rho}_{XY,i})$ ranging from 0.24−0.72 with lags of $\tau = 0$-51 days (Fig. 4, open circles). Two clear outliers, at $\tau = 93$ and 114 days, were disregarded (not shown). For each point plotted in Fig. 4, the corresponding ENSO state for that year, taken from historical data, is indicated. Three of the five (and four of the ten) largest correlations were associated with strong El Niño years, each with a lag of two

days or less between rainfall and streamflow (Fig. 4). Rainfall events in the upper reaches of the FRB can have a delayed effect on streamflow at Hope of up to ~1 week (BC River Forecast Centre, 2012), while in years with $\tau = 1$ to 8 weeks, soil

moisture that is near field capacity might be responsible for successive overland flow following rainfall events. However, Fig. 4 shows that these situations tend to occur during weak or moderate La Niña conditions.

Finally, we note the possibility that significant multi-day rainfall events coinciding with frozen or saturated soils might lead to occasional flooding at local scales within the FRB. While suitable soil moisture observations are not readily available at this time (see Sec. 5), we employ the VIC model to examine the role of soil moisture in both the FRB and its sub-basins using the regression framework in Section 3.4.

### 3.3.4 Relationships amongst observed predictors

Finally, we investigated relationships amongst observational variables unrelated to streamflow revealed in the correlogram of Fig. 3b. April-June mean temperature is positively correlated with spring warming rate and negatively correlated with 1 April SWE, both intuitively reasonable results, as is the positive correlation of FDD with SWE. Strong inverse relationships also exist between spring rainfall and: (i) $T_{amj}$ ($\hat{\rho} = -0.48$), and (ii) warming rate ($\hat{\rho} = -0.69$). Both of these relations are reasonable given that the rainfall fraction diminishes approximately linearly with decreasing temperature toward 0 °C, according to the Dai (2008) parameterization used here. Finally, cold season (Oct-Mar) rainfall is anti-correlated with FDD ($\hat{\rho} = -0.32$) (Fig. 3b).

### 3.3.5 Multivariate regression

The above results were derived using univariate linear regression and correlations computed using the Spearman rank method. However, we also would like to attribute interannual variance in the APF to the combined variances of the predictors. As described in Sec. 2.3, we constructed a number of MLR relations including all possible combinations of the above variables that showed a significant Spearman $\hat{\rho}$ when regressed individually against APF.

The MLR procedure yields the following multilinear regression fits:

$$\widehat{APF}_{obs} \text{ (m}^3 \text{ s}^{-1}) = 4535 + 11.40 \text{ SWE}_{Apr1} + 17068 \text{ } (dT/dt) - 603.2 \text{ } T_{amj} \qquad (R_{adj}^2 = 0.63, p < 10^{-9}) \qquad (4)$$

$$\widehat{APF}_{obs} \text{ (m}^3 \text{ s}^{-1}) = 4106 + 10.48 \text{ SWE}_{Apr1} + 17815 \text{ } (dT/dt) - 494.3 \text{ } T_{amj} - 402.3 \text{ NINO3.4} \qquad (R_{adj}^2 = 0.65, p < 10^{-9}) \qquad (5)$$

where the units of the predictors on the right-hand sides of Eq. (4) and (5) are provided in Table 2. Eq. (4) includes only local, basin-averaged predictors, while Eq. (5) includes the non-local influences of ENSO and PDO. Interestingly, only NINO3.4, not the PDO index, is a significant predictor of APF in the MLR, despite the comparable importance of both indices in the univariate regressions (Sec. 3.2). The residuals from both fits are indistinguishable from a Gaussian distribution at the 5% significance level, according to the Kolmogorov-Smirnov statistic. Additional parameters of the fits are summarized in Table 4. The predictors on the right-hand side of Eqs. (4) and (5) are ordered from left to right by decreasing partial $F$ value: e.g., SWE$_{Apr1}$ contributes the majority of the interannual variance in APF, followed by $dT/dt$, $T_{amj}$

and NINO3.4 (Table 4). Together, these variables explain 65% of the variance, 2% more than if the influence of ENSO is ignored. As mentioned above, rainfall variables do not contribute significantly to the interannual variability of the observed APF at Fraser-Hope.

**3.4 Observed versus VIC-simulated streamflow and SWE**

Before exploring relationships between APF and the wider array of variables available in the VIC hydrological model, as an evaluation exercise we compare several features of the VIC simulations with available observations in the FRB. In Fig. 5a, we compare the ten largest APFs in observations and VIC at Fraser-Hope station over the simulation period of 1955-2004. The APFs have been ranked by their observed magnitude, with the highest flow years at the left-hand side of the bar graph.
In most years, the VIC-simulated APFs are close to the observed values. Fig. 5b compares the daily climatology and interannual variability of VIC and observations over the same period. VIC tends to overestimate the magnitude of the APF by ~8% (multi-year mean) to 16% (multi-year maximum), and also simulates an APD ~5 days later than in the observations. VIC underestimates interannual streamflow variability over most of the year, except over the period of peak flow from June to mid-August, when it displays higher variability. However, given that the interannual coefficient of variation in observed
APF ($CV = \sigma/\text{mean}[APF]$) is 18%, that the VIC-simulated CV = 20%, and their degree of overlap, we conclude that the hydrographs are not significantly different.

Fig. 5c shows quantile plots of APF and APD for VIC compared to observations over the 1955-2004 period. Here, the permutation test was applied using the test statistic $S = [\text{mean}(APF)]_{VIC}/[\text{mean}(APF)]_{OBS}$ (Sec. 2.3). Fig. 5c shows that for APF, the VIC-simulated APF is indistinguishable from the stream gauge observations at the 5% significance level (i.e., $p$
$= 0.07 > 0.05$; $p = 0.24$ if the two outliers with VIC-simulated APF > 13,000 $m^3\,s^{-1}$ are removed), while the late bias in VIC-simulated APD is significant at the ~1% level. As our main concern in this work is with identifying key predictors of the APF, with less emphasis on APD, we conclude that VIC simulates APF reasonably well compared to observations.

Due to the dominant influence of annual snowpack on APF, it is also desirable to compare the VIC-simulated SWE with observed 1 April SWE measurements. In the analysis of the VIC simulation, we use $SWE_{max}$ as a predictor instead of
$SWE_{Apr1}$ since it is expected to be more closely linked to APF on physical grounds. In Fig. 5d, the annual cycle of the corrected, VIC-simulated SWE (Sec. 2.2) is compared with the observed 1 April SWE, with both quantities represented by their basin averages. While there is significant overlap in the interannual ranges of simulated and observed SWE, VIC appears to systematically underestimate the snowpack amount. However, it is important to recognize the sampling bias inherent in the observed, basin-average SWE: since the MSS stations tend to be located at high elevation (range: 750-2200
m), the observed estimates are likely biased high compared to a true basin average, as represented in VIC. In addition, we note that the apparent VIC biases in peak streamflow and SWE over the FRB are qualitatively consistent with VIC simulations over other nival basins in the Pacific Northwest (Salathé et al., 2014).

### 3.5 Relationships amongst VIC-simulated variables

In this section we apply the methodology of Sec. 3.3 to probe relationships amongst the wider array of variables available in the VIC hydrological model over the FRB. We begin with an analysis of the routed model streamflow at Fraser-Hope, as a representative outlet of the entire FRB, before examining VIC-modelled discharge at the outlets of four subbasins within the FRB in Section 3.6.

In addition to the PCIC-OBS variables used in the regression analysis of Sec. 3.3—which were used as daily forcings for the VIC simulation—we include a number of variables available in the model but not in observations, namely: annual maximum of SWE averaged over the FRB, $SWE_{max}$; calendar date of $SWE_{max}$; snowmelt rate, $d(SWE)/dt$; melt season length, $SWE_{len}$; and antecedent total column soil moisture (detailed definitions are given in Table 2). The regression results are again summarized as a correlogram (Fig. 6) and in Table 4. The annual snowmelt rate, $d(SWE)/dt$, was calculated as the best-fit, least squares linear slope of SWE between the dates of $SWE_{max}$ and APF. Formally speaking, $d(SWE)/dt$ is the snow ablation rate, which in VIC includes snowmelt as the dominant contribution along with evaporation and rain-on-snow (other snow removal processes occurring in nature such as blowing snow, avalanches, etc. are not simulated). For convenience, however, we refer to $d(SWE)/dt$ as simply the snowmelt rate with the understanding that these other processes may make minor contributions to snowpack disintegration.

### 3.5.1 Influence of large-scale climate modes on streamflow at Fraser-Hope

Through the influence of the forcing variables, the PDO and ENSO may affect the VIC-simulated APF at Fraser-Hope. Indeed, we find a slightly more robust, but qualitatively similar, relationship between the large-scale climate indices and modelled APF than for observed APF: as seen in Fig. 6, the correlations are Spearman $\widehat{\boldsymbol{\rho}} = -0.41$ for NINO3.4 and $\widehat{\boldsymbol{\rho}} = -0.38$ for PDO (both at $p < 10^{-2}$).

### 3.5.2 Snow water equivalent and snowmelt

As noted for the observed 1 April SWE, we find that $SWE_{max}$ exercises a strong control on APF ($\hat{\rho} = 0.70$, $p < 0.001$; Fig. 6). The latter lags $SWE_{max}$ by an average of 84 days (range 52-127 days) over the simulation period; examples for specific years are displayed in Fig. 9. Further, Table 5 shows that of the top ten APF years in this period, seven were in the top ten of $SWE_{max}$. Yet, the highest APF corresponds to the 6th-largest $SWE_{max}$ (1982), while the 4th-highest APF occurred in an average year for snow accumulation (1958). It is therefore of interest to investigate what conditions, independent of snow accumulation, led to the comparatively large APF in those years.

Snowmelt displays significant positive correlations with APF ($\hat{\rho} = 0.43$) and several other predictors, namely: $SWE_{max}$ ($\hat{\rho} = 0.63$), FDD ($\hat{\rho} = 0.38$), date of $SWE_{max}$ ($\hat{\rho} = 0.32$), $dT/dt$ ($\hat{\rho} = 0.25$), and maximum annual soil moisture ($\hat{\rho} = 0.64$). In years with high $SWE_{max}$, snowmelt in spring recharges soil moisture (over unfrozen ground; see Sec. 3.5.3), sometimes to field capacity, which leads to increased runoff and the strong positive correlations between snowmelt, soil

moisture and APF. Significant negative correlations are also found with melt season length ($\hat{\rho} = -0.53$) and the NINO3.4 and PDO indices ($\hat{\rho} = -0.28$ and $-0.46$, respectively). The inverse relationship with the large-scale indices should be considered in light of their similar relationship with $SWE_{max}$ ($\hat{\rho} = -0.47$ and $-0.62$, for NINO3.4 and PDO respectively): higher snowpack accumulates during La Niña and negative PDO phases, with reduced snowmelt due to cooler temperatures.

### 3.5.3 Soil moisture

In our VIC simulation, the basin-averaged annual peak soil moisture, $SM_{max}$, occurs during the snowmelt period from mid-March to mid-June, most often near the end of May, well after the day of $SWE_{max}$ and approximately three weeks in advance of the APF (for an illustration depicting individual years, see Fig. 9). The direct influence of both $SWE_{max}$ ($\hat{\rho} = 0.83$) and snowmelt ($\hat{\rho} = 0.64$) on $SM_{max}$ is detected at high confidence ($p < 10^{-3}$), while $SM_{max}$ is, in turn, strongly correlated with APF ($\hat{\rho} = 0.65$, $p < 10^{-3}$) in the annual time series (Fig. 6). A cross-correlation analysis of the respective daily time series over the entire common period reveals that, on average, SWE leads SM by 68 days (maximum cross-correlation, $R = 0.65$), SM leads streamflow by 22 days ($R = 0.85$), and SWE leads streamflow by 105 days ($R = 0.74$). The higher cross-correlation between SM and streamflow indicates that the annual cycle of SM is a better predictor of the daily streamflow hydrograph than SWE, despite the above-mentioned superiority of $SWE_{max}$ as an annual predictor of APF. This is a reasonable result, since during the freshet, daily SM integrates contributions from both snowmelt and precipitation. On the other hand, $SM_{max}$ could be considered inferior to $SWE_{max}$ as a predictor of APF with respect to its much shorter lead time. $SM_{max}$ also exhibits significant relationships with FDD ($\hat{\rho} = 0.33$), $T_{amj}$ ($\hat{\rho} = -0.27$), and the NINO3.4 and PDO indices ($\hat{\rho} = -0.44$ for both). The negative phases of ENSO and the PDO bring more rain and snow, which consequently enhances soil moisture.

Also of interest is a possible relationship between APF and SM preceding the snowmelt period, for example, in fall before snow accumulation begins. Numerous studies point to the influence of antecedent soil moisture on seasonal streamflow in nival catchments (Maurer and Lettenmaier, 2003; Berg and Mulroy, 2006; Williams et al., 2009; Harpold and Molotch, 2015). Using a suite of land surface models including VIC, Koster et al. (2010) and Mahanama et al. (2012) demonstrated improvement of March-July streamflow forecast skill over the western United States using antecedent (January 1) soil moisture in addition to snow amount as a predictor. The basic mechanism is that antecedent reduced storage capacity of wet or frozen soils leads to more of the snowmelt being routed to discharge during the spring freshet, and vice-versa for dry soils. To investigate the interannual sensitivity of APF to antecedent SM, we used monthly mean SM from the preceding August through November in turn as predictors of APF, but found no significant correlations. This insensitivity may be due to the lack of a frozen soil parameterization in the version of VIC used here. In this implementation, soil drainage over the cold season tends to be overestimated (Cherkauer and Lettenmaier, 2003), which could cause a decoupling between autumn SM and spring freshet flows.

### 3.5.4 Temperature and rainfall

As in the observations, FDD and spring warming rate $dT/dt$ are positively correlated with APF ($\hat{\rho} = 0.24$ and $\hat{\rho} = 0.38$, respectively; Fig. 6). The connection between $dT/dt$ and APF is stronger than in the observations but, interestingly, there is no significant correlation between $T_{amj}$ and APF as exists in the observational data (Eq. [4] and [5]). Further, the results in Table 5 suggest that large APFs in average SWE years are characterized by large $dT/dt$ (e.g., 1982, 1958, 2002).

Again as in the observations (Sec 3.3.3), no significant correlations are found between any of the rainfall measures and VIC-simulated APF. However, as also found in the observations, there are indications of a rainfall-streamflow connection at other times of the year. Repeating the cross-correlation analysis between streamflow and CRI as in Sec. 3.3.3 reveals that 20 of 50 years exhibit a significant relationship at some time during the year, with $\max(\hat{\rho}_{XY,i})$ ranging from 0.27−0.66 with lags of $\tau = 2\text{-}55$ days. Two clear outliers, at $\tau = 91$ and 142 days, were disregarded. Fig. 4 (triangles) shows that most (nine) of the rainfall-influenced years occur during the El Niño phase, while four occur during La Niña and seven during neutral years. Twelve of the 20 rainfall-influenced years occur during the cool phase of the PDO. More often than not, however, CRI lags daily streamflow, suggesting little to no relationship. With regard to other relationships, we find that cold season rainfall is correlated with spring rainfall ($\hat{\rho} = 0.32$) and anti-correlated with FDD ($\hat{\rho} = -0.32$) (Fig. 6). As mentioned in Sec. 3.3.4, spring rainfall is negatively correlated with $T_{amj}$, $dT/dt$, and snowmelt, but with slightly different $\hat{\rho}$ values due to the different lengths of averaging period in the calculations compared to the observed case (as per the definitions in Table 2).

### 3.5.5 Multivariate regression

As in the case of the observed variables, we constructed MLR relations including all combinations of the variables in Fig. 6 that showed a significant Spearman $\hat{\rho}$ and selected the optimal MLR based on the criteria specified in Sec. 3.3.5. One predictor that was excluded from the MLR was $SM_{max}$: its high correlation with $SWE_{max}$ indicates that its independent explanatory power is limited (Sec. 3.5.3). The resulting relationship is:

$$\widehat{APF}_{VIC} \ (\text{m}^3 \ \text{s}^{-1}) = 3239 + 37.14 \ SWE_{max} + 26842 \ (dT/dt) - 57.69 \ SWE_{len} + 1770 \ d(SWE)/dt \quad (R_{adj}^2 = 0.75, p < 10^{-9}) \quad (6)$$

In contrast to the MLR constructed for observed streamflow, Eq. (5), neither of the large-scale climate indices has a significant influence, nor is $T_{amj}$ an important predictor. Nevertheless, the four variables on the right-hand side of Eq. (6) account for 75% of the interannual variance in APF. Three of the four were identified in the univariate regressions as important, with the exception being the length of the melt season, $SWE_{len}$. The latter displays a strong anti-correlation with snowmelt ($\hat{\rho} = -0.55$, Fig. 6), suggesting that the two variables are not independent. However, since removing each of $SWE_{len}$ and $d(SWE)/dt$ from the MLR in turn yields a significantly poorer fit ($R_{adj}^2 = 0.71$ for $d(SWE)/dt$ only and $R_{adj}^2 = 0.69$ for $SWE_{len}$ only), it seems that both predictors have some explanatory value. The absence of both the PDO and

NINO3.4 indices from the MLR could be a reflection of the limitations of the driving data, inasmuch as the processing of gridded station data to the regular grid may weaken the influence of the large-scale climate drivers (Sec. 2.1).

**3.6 Relationships amongst VIC-simulated variables at the subbasin scale**

The same analysis as conducted for the entire FRB was repeated for each of the four subbasins comprising nearly 70% of the
annual flow at Fraser-Hope (Table 1; Kang et al. 2016). The results of the univariate regressions are presented as a correlogram in Fig. 7 while those for the MLR are provided in Table 4.

Overall, the relationships in the subbasins mirror those seen in the FRB as a whole. The univariate analysis shows that in all four subbasins, $SWE_{max}$ and $SM_{max}$ are good predictors of APF, while snowmelt and FDD exhibit positive correlations with APF in three of the four subbasins, Chilko being the exception (Fig. 7). While $SWE_{max}$ has a slightly
stronger influence on APF than $SM_{max}$ in the FRB, $SM_{max}$ is more influential at the subbasin scale, explaining over 75% of the variance in APF in Quesnel (compared to 40% for $SWE_{max}$). The correlation between spring $dT/dt$ and APF is strong in Thompson-Nicola ($\hat{\rho} = 0.50$, $p < 0.05$) but weaker in Quesnel ($\hat{\rho} = 0.25$, $p < 0.1$) and not significant in the other two catchments. Three of the four subbasins exhibit a strong inverse relationship between APF and the NINO3.4 and PDO indices, again with the exception of Chilko. The weak dependence of streamflow on the PDO, SOI and Pacific-North
American indices in this and other catchments in the western FRB was also noted by Thorne and Woo (2011), who attributed this insensitivity to the low magnitude of discharge in these tributaries. These authors speculated that the main trunk of the Fraser, by contrast, integrates the influence of regional climate forcings on upstream catchments, bolstering the teleconnections. The insensitivity of APF to the local, basin-averaged predictors might be related to the smaller size of the Chilko compared to the other three subbasins, which lowers the signal-to-noise ratio of the basin averages. This, along with
the comparatively lower APFs at the basin outlet, makes the detection of significant correlations more challenging in Chilko compared with the other subbasins.

Moving now to the MLR analysis, the results in Table 4 demonstrate that in all four subbasins, $SWE_{max}$ is the most skillful predictor of APF, again in agreement with the results for the FRB as a whole. Likewise, in three of the four catchments, $dT/dt$ emerges as the next most significant variable. The exception is the Upper Fraser basin, where $SWE_{len}$ is
the second most skillful predictor. In this subbasin, APF is inversely correlated with $SWE_{len}$ in the MLR, while the latter is in turn anti-correlated with both $dT/dt$ ($\hat{\rho} = -0.81$, $p < 0.05$) and $T_{amj}$ ($\hat{\rho} = -0.25$, $p < 0.1$) in the univariate regressions (Fig. 6). These results suggest that anomalously warm springtime temperatures (compared to the preceding winter) are at the heart of the APF-$SWE_{len}$ relationship in the Upper Fraser, not unlike what is seen in the other subbasins.

In contrast to the FRB as a whole, rainfall does appear to have a weak influence on APF in two of the subbasins.
However, in each subbasin, APF is sensitive to a different measure of rainfall. In Thompson-Nicola, the basin with the largest area and second-highest mean elevation, $R_{Oct-Mar}$ is positively correlated with APF, while in the Upper Fraser, $R_{Spring}$ displays a somewhat weaker positive relationship to APF. In the Quesnel, a smaller basin of lower mean elevation, the preceding September mean soil moisture is an effective predictor of APF. Interestingly, neither the NINO3.4 nor the PDO

index is an effective predictor of APF in the MLR of any of the subbasins, despite the fact that strong inverse relationships are still seen in the univariate correlogram (Fig. 6).

### 3.7 Co-dependence of streamflow predictors

The above results make clear that, of the predictors considered in the FRB and its subbasins, APF is primarily influenced by SWE$_{max}$ and the rate of warming in spring, $dT/dt$. To explore the co-dependence between predictors, we show in Fig. 8 a scatterplot of the relative anomaly of each of these two predictors, $\Delta X/\overline{X} = (X_i - \overline{X})/\overline{X}$, where $X_i$ is the value of predictor $X$ at year $i$ and $\overline{X}$ is the long-term mean (1955-2004), with the corresponding streamflow relative anomaly $\Delta Q/\overline{Q}$ indicated by the point colour. This type of plot has been used previously to explore the elasticity (i.e., non-linearity) of streamflow as a function of covariates (e.g., Andréassian et al., 2016). However, here we employ it primarily as an additional illustration of the co-dependencies identified by the MLR analysis.

In the FRB, the roughly uniform pattern of scatter in the vertical and horizontal directions indicates a weak co-dependence between $\Delta$SWE$_{max}$ and $\Delta(dT/dt)$, consistent with expectation—$dT/dt$ is computed only after the day of SWE$_{max}$ each year—and with the results of Figs. 5 and 6. The corresponding $\Delta Q/\overline{Q}$ values display an overall gradient from bottom left to top right, i.e. from lower than average SWE$_{max}$ and $dT/dt$ to higher than average SWE$_{max}$ and $dT/dt$, again consistent with the univariate and multivariate regression results. Qualitatively similar results are found in the Chilko, Thompson-Nicola and Quesnel subbasins.

In the Upper Fraser subbasin, the primary predictor remains SWE$_{max}$ but the secondary predictor is SWE$_{len}$. As a result, the scatterplot displays a weak co-dependence between relative changes in the two variables, with some clustering toward the diagonal. Furthermore, the overall gradient in $\Delta Q/\overline{Q}$ is from top left to bottom right, i.e. from lower than average SWE$_{max}$ and higher than average SWE$_{len}$ to higher than average SWE$_{max}$ and lower than average SWE$_{len}$. That is, in years when an anomalously large snowpack ($\Delta$SWE$_{max}$ > 0) melts fairly quickly ($\Delta$SWE$_{len}$ < 0), streamflow tends to be larger than usual ($\Delta Q$ > 0).

### 4 Case studies: High-flow years in the FRB

While the MLR approach captures the relationship between various predictors and APF in the FRB and its subbasins in a statistical manner, it can miss unusual weather factors that are important in specific years. Both historical data and anecdotal accounts of large historical floods in the region place a large emphasis on the precise seasonal development of warming temperatures and rainfall during the snowmelt period from early April to late June as being a critical factor in the development of damaging floods (BC River Forecast Centre, 2012; Septer, 2007). For this reason, we present in this section an "anatomy" of streamflow and the associated key hydrological variables in years of particularly high VIC-simulated

discharge at the Fraser-Hope station. We focus on VIC-simulated variables since full time series of all variables of interest are available to construct both a climatology and annual cycles for specific years.

Table 5 shows the top ten APFs occurring in the VIC simulation from 1955-2004, with corresponding rankings of other key predictors entering the regression analysis of Sec. 3. Seven of the top ten $SWE_{max}$ years are in this group. This highlights the dominant effect of spring snowpack on APF magnitude in the FRB. A similar result holds for observed basin-averaged 1 April SWE and APF at Fraser-Hope: eight of the top ten 1 April SWE years are also amongst the top ten observed APFs. Looking at results for the other predictors, seven of the top ten $SM_{max}$ years (not surprising given the linkage between $SWE_{max}$ and $SM_{max}$ noted in Sec. 3.4.3), four of the top ten spring warming rates and three of the top ten FDDs are in the group of top ten APFs. Only one of the top ten spring rainfalls is in the group, which happens to be the highest simulated rainfall in 1999 (8th largest APF). These results are therefore consistent with the key predictors identified for APF in the regression analysis of Sec. 3.

Fig. 9 shows the annual cycle of discharge at Fraser-Hope along with other variables of interest over the calendar year for three of the top ten APF years: 1958, 1972 and 1999. The daily evolution of discharge $Q$ at Fraser-Hope and basin-averaged SWE, $T$, CRI and SM are plotted, along with their 1955-2004 climatologies, as an aid to evaluating how anomalous a given year is. A 15-day smoothing filter was applied to $T$ to better highlight sub-seasonal trends. Also shown in the bottom two subpanels of each panel are the difference between the slope of $T$ calculated over a 61-day moving window in the year of interest and its climatology (the warming rate anomaly), and the difference of the snowmelt rate from its climatology calculated over a 15-day moving window. Both anomalies were set to zero if $T < 0$, since we are interested in behaviour during the melt season only. We consider each year in turn, as each exhibits a unique phenomenology that illuminates the development of the APF in that year. Also, in this section we provide descriptions of the two largest flooding events ever recorded in the FRB—the freshets of 1894 and 1948—both of which preceed the simulated period. It is instructive that there are similarities between the meteorological precursors and covariates in those years and the explicitly simulated years highlighted below.

**4.1 Warm spring in an average SWE year: 1958**

APF in the year 1958 ranked fourth over the 1955-2004 period of the simulation (Table 5) and 13th in observed APF. Yet, as simulated by VIC, 1958 was a normal SWE year, with $SWE_{max}$ achieved near the end of March. Rainfall and soil moisture were also near normal up to this point, while winter temperatures were above normal but still below freezing (Fig. 9a). At the beginning of the melt season in April, $dT/dt$ began to exceed its climatological value, prompting rapid snowmelt toward the end of that month. The anomalous snowmelt persisted until the end of May, about the time of the APF (APD), while the period of elevated warming rate continued into early June. The snowmelt pulse produced a coincident soil moisture anomaly, which is remarkable in that these quantities are averaged over the entire FRB, yet their behaviour is consistent with small-scale processes. Subsequent to the APD, the soil moisture anomaly became negative, apparently in response to the prolonged warming, which would lead to enhanced evaporation, and lower than normal rainfall over the

summer. Fig. 9a demonstrates that neither rainfall nor high soil moisture prior to freeze-up were pivotal factors in producing the anomalously large APF in this year. Indeed, rainfall amounts were well below normal between snowmelt initiation and the APD (49th out of 50 years; Table 5).

## 4.2 A high-SWE year: 1972

The second-highest simulated APF over the 1955-2004 period occurred in 1972, which also featured the second-highest $SWE_{max}$ (Table 5). This year also exhibited the largest observed APF over the same period (Fig. 5a). While the preceding winter was colder than normal—evidently a factor in generating an anomalously large snowpack—the spring warming rate was above-normal, resulting in a strong snowmelt anomaly lasting from May until just after the APD in early June (Fig. 9b). As in 1958, this snowmelt pulse generated a large soil moisture anomaly, which likely produced more overland flow during the freshet. This evidence suggests that the extreme APF in 1972 was primarily caused by a combination of elevated warming in spring and a heavy snowpack, which provided a large water surplus during the freshet. However, there is also an indication that a series of heavy rainfall episodes starting in early June, just days before the APD, affected the APF and particularly the duration of the high-flow period. An extended period of anomalously large rainfall is seen in the CRI between early June and late July, with a clear response in SM and in the persistence of the high-discharge anomaly until the end of August—both are clearly distinguishable from their counterparts in 1958 (Fig. 9a).

Finally, it is worth noting that 1982, the year with the highest simulated APF (not shown in Fig. 9), was characterized by a very similar evolution of predictors as in 1972, including an extended, coincident period of anomalously large rainfall. However, in that case, there were two additional elements: 1) the spring warming rate was the highest of any year in the record (Table 5), and; 2) a strong warm anomaly ~+5 °C occurred in mid-June just prior to the APD, which evidently generated enough additional snowmelt runoff to position that APF as the highest over the simulated period.

## 4.3 Influence of rainfall in a high-SWE year: 1999

The year 1999 holds the distinction of having the highest simulated spring rainfall in the record. It also had the fourth-largest VIC-simulated $SWE_{max}$, while ranking eighth in APF (fifth in observed APF; see Fig. 5a and Table 5). The temperature development was unremarkable, albeit a little colder than the climatology in the freshet months of April and June (Fig. 9c). The hydrograph reflects this increased role of rainfall by its evident synoptic time scale variability, in contrast to its smoother counterpart in the snowmelt-dominated years examined above (Fig. 9a,b). Indeed, the character of the hydrograph over the freshet period is such that isolating the single largest APF misses key features of the flow development, which involves not one but four peak flows of roughly equal magnitude spanning the period from mid-June to mid-August. Consequently, the use of annual predictors with a single APF as predictand, as adopted in our univariate and multivariate regressions, is likely to underweight a strong influence of rainfall in a particular year. Table 3 shows that the maximum cross-correlation of deseasonalized daily streamflow and CRI anomalies during 1999 is 0.61 at a lag of 35 days (CRI leading streamflow), while in 1958 the maximum is just 0.38 at a lag of 47 days. This implies that rainfall should be treated

differently than SWE and other predictors to correctly capture its influence on APF. The evolution of SM is also different than in the prior cases considered, insofar as it displays step-like jumps in mid-March, early April, and mid-May followed by a plateau at a high level until early July (Fig. 9c). This SM anomaly persisted for the remainder of the year, except for a brief two-week period in November. While in snowmelt-dominated years SWE approaches its climatological value in late summer
(except in 1958, when it fell below that value), in 1999 a positive SWE anomaly persisted throughout the year.

### 4.4 The two highest flow years in the FRB: 1894 and 1948

A description of meteorological precursors to the 1894 and 1948 floods is available from the BC MOE (BC MOE, 2008) and also the historical review, based on newspaper accounts, of Septer (2007). According to limited archival data
available at two interior locations in the FRB, snowfall amounts during the winter of 1893-94 were near normal over the FRB as a whole, although the snowpack remaining from the previous summer was reportedly larger than usual. While the spring of 1894 was cold and wet in the FRB, temperatures rose rapidly in late May, becoming unseasonably warm near the end of that month. By the end of May, the Fraser overtopped its banks at many locations and existing dikes were breached, leading to the "Great Chilliwack Flood" in the densely populated Lower Fraser valley. The Fraser River at Mission (near
Hope) peaked on 5 June, but floodwaters did not subside in many areas until early July. A hydraulic modelling exercise conducted in 2008 estimated a peak discharge at Hope in the range of 16,000 to 18,000 $m^3$ $s^{-1}$, with a best guess of 17,000 $m^3$ $s^{-1}$ (BC MOE, 2008).

The meteorological record for the FRB in 1948 is considerably more complete than for 1894. It shows that precipitation was well above normal for a long period in advance of the 1948 freshet, with excess rainfall in autumn 1947
and heavy snowfall in the subsequent winter and early spring. As a result, soils in many areas were likely near saturation at the time of freeze-up, an important factor for the partitioning of runoff from spring snowmelt. Also, the accumulated reservoir of snow was larger than normal. The early spring of 1948 was cooler than average to 20 May, followed by a rapid increase to unseasonable warmth in late May-June. Thus, sensible heat inputs to the existing high snowpack in the FRB were large, leading to a rapid and voluminous release of meltwater to lower elevations. The peak flow at Hope on 31 May was
measured at 15,200 $m^3s^{-1}$, which remains the highest value in the instrumental record (Septer, 2007; BC MOE, 2008). Apparently, neither the 1894 nor the 1948 event was characterized by excessive rainfall coincident with the meltwater peaks.

### 5 Discussion and Conclusions

In Sec. 3.4, we showed that the VIC model driven by gridded observations provides an adequate simulation of streamflow
and maximum annual SWE in the FRB. The mutual resemblance of the correlograms (Figs. 3b and 6), which summarize the univariate linear regression fits to observed and VIC data, along with the similar forms of the respective MLR models (Eqs. [1]-[3]), give one further confidence in the ability of the VIC model to simulate interactions between the key controls of streamflow acting in the real system. Furthermore, differences in these controls in the observed and modelled cases suggest

fruitful avenues of further research. For example, the use of more complete snow survey or satellite products might permit the estimation of $SWE_{len}$ or snowmelt rate, and confirmation of the influence of these terms appearing in the VIC-derived MLR, Eq. (6). Or conversely, improved driving data for the VIC model might reveal the influence of the PDO and/or ENSO explicitly in the MLR, or the inverse correlation with spring temperature and/or freezing degree days. The relationships derived between the basin-averaged predictors examined and streamflow at the major basin and subbasin outlets in the FRB should be relevant to other large, nival watersheds elsewhere on the globe, since the physical drivers (e.g., temperature seasonality and precipitation phase change) are generic and scale-independent. However, it is important to recognize that the influence of large-scale drivers (e.g., ENSO, PDO) will differ depending on the specific geographic setting.

Interestingly, in the historical climate record we find little prospect of using basin-averaged, rainfall-related indices as predictors of APF at Fraser-Hope. It is possible, however, that at subbasin scales, and/or in combination with an appropriate soil moisture indicator, some measure of rainfall might prove useful for predicting localized flooding. This has been a valuable strategy for APF/APD hindcasting in nival-pluvial and pluvial catchments (Neiman et al., 2011; Surfleet and Tullos, 2013). Moreover, given that the rain-to-snow ratio will increase in the FRB in response to further warming (see below), future studies of projected streamflow could uncover an explicit rainfall-streamflow connection at the basin scale.

We emphasize that the identification of the principal controls on APF at the interannual time scale within the MLR framework has its limitations. The relationships derived from historical inputs can only be considered broadly indicative of expected behaviour, and then only in a basin-averaged sense. Although based on only 50 years of simulated basin hydrology driven by observed data, the VIC results imply that a high $SWE_{max}$, say in the top quartile of historical values (i.e. seven of the top ten in Table 5), increases the chance of an upper quartile APF. The long interval between $SWE_{max}$ and APF, of order 2-4 months, makes the former a valuable early warning indicator of possible flooding in the lower FRB (BC River Forecast Centre, 2012). However, it is also true that three of the top ten $SWE_{max}$ years (1976, ranked 3rd; 1991, 8th; and 1971, 9th) did not exhibit remarkable floods: in those years, the corresponding APFs ranked 20th, 30th and 23rd. And while $SM_{max}$ has about the same success rate as a predictor of APF—seven of the top ten APF years are top ten $SM_{max}$ years—these are the same as for $SWE_{max}$, meaning that soil moisture during the freshet is of little additional predictive value. As mentioned in Sec. 3.5.3, the lack of a connection between fall SM and APF in the historical VIC simulation suggests that future model studies should include the effect of soil freezing to more realistically simulate the spring thaw. Further, the estimation of groundwater storage and its evolution using NASA's Gravity Recovery and Climate Experiment (GRACE), which has proved promising for flood forecasting in large, snowmelt-dominated basins (Reager et al., 2014; Wang and Russell, 2016), could find useful application in the FRB.

In Section 3, we demonstrated that the springtime warming rate, $dT/dt$, was the next most skillful predictor of APF in the FRB. Moreover, $dT/dt$ appears to provide additional predictive value over that of $SWE_{max}$ and $SM_{max}$, insofar as two of the top ten $dT/dt$ years are not top ten $SWE_{max}$ and $SM_{max}$ (1958 and 2002; Table 5). Thus, the combination of an upper quartile $SWE_{max}$ with a high $dT/dt$ over the snowmelt period may presage flooding at Fraser-Hope station (nine out of the top ten APFs were characterized by one or the other; four featured both). The definition of $dT/dt$ used here is dependent upon the

APD, which is of course unknown at the time of $SWE_{max}$; however, in a predictive context, one could simply use the most recent daily $T$ observation to compute the warming rate within a window of increasing duration. Better still, if hydrological forecasting on the daily to seasonal time scale is of paramount interest, employing a forecast methodology based on an ensemble of hydrological simulations and multiple initializations would constitute a superior approach (Bazile et al., 2017).

5        In the event that future simulations of the FRB were to furnish annual time series of the basin-averaged predictors appearing in Eq. (6) and/or Table 4—for example from a coupled global climate model—could the MLR relationships derived in this paper be used for predictions of future APF? We could answer in the affirmative if both of the following conditions were to hold: 1) the dominant predictors of APF in the late twentieth century (defining a multi-decadal reference period) remain unchanged in the future period of interest, and; 2) the APF predictions comprise multi-year or decadal scale

anomalies from the reference period: e.g., APF in 2080-2100 compared to 1980-2000. Assumption 1) is placed in some doubt due to the significant warming projected for the FRB in future (Shrestha et al., 2012), which will increase the rain-to-snow ratio over large areas of the basin already characterized by an annual mean temperature near 0 °C. This suggests that rainfall may become a key predictor for high streamflow in the region, with perhaps a less pivotal role played by $SWE_{max}$. Regarding condition 2), the suppression of interannual variability in the MLR fits underestimates interannual variations in

APF, meaning that only decadal-scale averages of the predicted APF would be meaningful.

        Of what use, then, are the predictor sets derived from the MLR analysis? The application of nonstationary extreme value theory to APFs suggest at least one promising avenue of progress (Towler et al., 2010; Shrestha et al., 2017). In such models, the parameters of fitted distributions (e.g., location, scale and shape parameters of the generalized extreme value (GEV) distribution) are taken to be functions of climate covariates. Temperature and precipitation are often chosen for this

purpose, due to their availability in observations and climate models and the presumption that they should influence streamflow at least under some circumstances. But as we have shown, in a large nival basin such as the FRB, rainfall has a weak influence on APF (although it would be expected to influence smaller magnitude flows), while the warming rate is more influential than temperature itself in determining APF magnitude. Hence, it may be worth introducing snowfall or $SWE_{max}$ and/or spring $dT/dt$ as covariates of the GEV parameters in the fitting of nonstationary probability distributions of

APFs. We leave this as an interesting topic for future investigation.

**Acknowledgements**

We thank Markus Schnorbus and Arelia (Werner) Schoeneberg for helpful discussions regarding the VIC model, Reza Najafi for providing observed SWE data for the Fraser basin, Faron Anslow for providing the regional temperature and precipitation time series data over BC, and J. Cunderlik for sharing station locations from Cunderlik and Ouarda (2009). The

comments of Siraj Ul Islam and referees S. Déry, M. Jenicek and an anonymous reviewer also led to substantial improvements to the final paper. CC is supported by the NSERC-funded Canadian Sea Ice and Snow Evolution (CanSISE) Network.

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

# Tables

Table 1: Water Survey of Canada (WSC) hydrometric stations and corresponding basin and subbasin characteristics.

| Station name (WSC ID) | Basin/subbasin name | Period of record | Subbasin Area [km$^2$] | Mean elevation [m] |
|---|---|---|---|---|
| Fraser R. at Hope (08MF005) | Fraser (FRB) | 1912-2014 | 217000 | 1330 |
| Fraser R. at Shelley (08KB001) | Upper Fraser | 1951-2014 | 32400 | 1308 |
| Quesnel R. near Quesnel (08KH006) | Quesnel | 1939-2013 | 11500 | 1173 |
| Thompson R. near Spences Bridge (08LF051) | Thompson-Nicola | 1952-2013 | 54900 | 1747 |
| Chilko R. near Redstone (08MA001) | Chilko | 1927-2013 | 6940 | 1756 |

Table 2: Predictor variables and descriptions including their origin. Acronyms are as defined in the text, with the exception of ESRL/GCOS = Earth System Resource Laboratory/Global Climate Observing System (National Oceanic and Atmospheric Administration, NOAA), and JISAO = Joint Institute for the Study of the Atmosphere and Ocean (University of Washington/NOAA).

| Predictor | Units | Description | Source |
|---|---|---|---|
| Maximum annual snow, $SWE_{max}$ ; 1 April SWE, $SWE_{Aprl}$ | mm | Annual maximum snow water equivalent (VIC); 1 April snow water equivalent (OBS) | VIC; MSS |
| October mean total column soil moisture | mm | Previous October mean value | VIC |
| Cold season rainfall | mm | Sum of rainfall between 1 Oct – 31 Mar | PCIC-OBS |
| Spring rainfall | mm | Sum of rainfall between days of $SWE_{max}$ (VIC) or $SWE_{Aprl}$ (OBS) and annual maximum streamflow (APF) | PCIC-OBS |
| APF rainfall | mm | Sum of rainfall from 15 days prior to 5 days after APF | PCIC-OBS |
| Freezing degree days | °C | Absolute value of sum of negative daily mean $T < 0$ °C from 1 Oct – 31 Mar | PCIC-OBS |
| Spring warming rate | °C day$^{-1}$ | Slope of daily mean $T$ between days of $SWE_{max}$ and $T_{max}$ | PCIC-OBS |
| Snowmelt rate | mm day$^{-1}$ | Slope of daily SWE between dates of $SWE_{max}$ and APF | VIC |
| Date of $SWE_{max}$, $t_{SWEmax}$ | day | Calendar day of maximum SWE | VIC |
| Melt season length, $SWE_{len}$ | days | Date of 0.25 $SWE_{max}$ minus date of $SWE_{max}$ | VIC |
| NINO3.4 index | °C | HadISST1 anomaly over 5 °N -5 °S and 170 °W -120 °W | ESRL/GCOS |
| PDO index | °C | Leading principal component of monthly SST anomalies in the North Pacific Ocean, poleward of 20 °N | JISAO |

Table 3: Trends in observed and VIC-simulated variables over the period of each record. Results are only shown for those variables that contain a statistically significant trend at the $p < 0.05$ level. The trend was calculated using the Theil-Sen median slope estimator (Sec. 2.3). Slightly different results are obtained for observed inputs to the VIC model due to the difference in start and end years of the time series. The residuals of each fit were checked for autocorrelation, with none detected.

| | Observations (1956-2006) | | VIC Model (1955-2004) | |
|---|---|---|---|---|
| **Variable (units of trend)** | **Trend (Sen)** | **Pearson $R^2$** | **Trend (Sen)** | **Pearson $R^2$** |
| APF (m$^{-3}$ s$^{-1}$ yr$^{-1}$) | −36.5 | 0.10 | – | – |
| Freezing degree days (°C day yr$^{-1}$) | −6.29 | 0.14 | −6.29 | 0.11 |
| April-June mean $T$ (°C day yr$^{-1}$) | 0.026 | 0.10 | – | – |
| Spring Rain (mm yr$^{-1}$) | – | – | 0.927 | 0.13 |
| PDO (yr$^{-1}$) | 0.035 | 0.16 | 0.038 | 0.17 |
| Snowmelt rate (mm day$^{-1}$ yr$^{-1}$) | n/a | n/a | −0.0137 | 0.10 |
| Melt season length (day yr$^{-1}$) | n/a | n/a | 0.225 | 0.12 |

Table 4: Fitted parameters from multilinear regression of the form: $Y = a_0 + a_1X_1 + a_2X_2 + a_3X_3 + ...$ . For each fit, all partial $p$-values are less than 0.05. For each of the subbasins, only VIC results are given, due to the small number of April 1 observed SWE values available. Results given for OBS include basin-averaged predictors over the FRB, while those for $OBS_{ext}$ include indices of large-scale climate variability.

| Basin / subbasin | Predictors | | | | Coefficients | | | | | Partial $F$ | | | | $F$ | $R^2_{adj}$ |
|---|---|---|---|---|---|---|---|---|---|---|---|---|---|---|---|
| | $X_1$ | $X_2$ | $X_3$ | $X_4$ | $a_0$ | $a_1$ | $a_2$ | $a_3$ | $a_4$ | $F_1$ | $F_2$ | $F_3$ | $F_4$ | | |
| FRB | | | | | | | | | | | | | | | |
| OBS | $SWE_{Apr1}$ | $dT/dt$ | $T_{amj}$ | | 4535 | 11.49 | 17068 | −603.2 | | 60.9 | 14.0 | 13.0 | | 29.3 | 0.63 |
| $OBS_{ext}$ | $SWE_{Apr1}$ | $dT/dt$ | $T_{amj}$ | NINO3.4 | 4106 | 10.48 | 17815 | −494.3 | −402.3 | 65.4 | 15.0 | 13.9 | 4.46 | 24.7 | 0.65 |
| VIC | $SWE_{max}$ | $dT/dt$ | $SWE_{len}$ | $d(SWE)/dt$ | 3239 | 37.14 | 26842 | −57.69 | 1769.6 | 89.9 | 31.8 | | | 57.3 | 0.70 |
| Upper Fraser | $SWE_{max}$ | $SWE_{len}$ | $R_{Spring}$ | | 2413 | 5.257 | −28.43 | 3.453 | | 39.0 | 16.9 | 2.64 | | 19.5 | 0.53 |
| Quesnel | $SWE_{max}$ | $dT/dt$ | $SM_{Sep}$ | | −542.0 | 1.834 | 988.3 | 2291.2 | | 55.1 | 19.0 | 5.77 | | 26.6 | 0.61 |
| Thompson-Nicola | $SWE_{max}$ | $dT/dt$ | $R_{Oct-Mar}$ | | −805.8 | 5.181 | 7199.8 | 4.341 | | 97.4 | 39.1 | 7.75 | | 48.1 | 0.75 |
| Chilko | $SWE_{max}$ | $dT/dt$ | | | −142.6 | 1.152 | 786.2 | | | 56.5 | 8.45 | | | 32.5 | 0.56 |

Table 5: Top ten APF years at Fraser-Hope in the VIC simulation over the period 1955-2004, along with corresponding ranks of other basin-averaged predictors. Years in boldface font are highlighted in Fig. 9.

| Rank APF | Year APF | Rank SWE$_{max}$ | Rank SM | Rank FDD | Rank $dT/dt$ | Rank Spring Rain |
|---|---|---|---|---|---|---|
| 1 | 1982 | 6 | 6 | 14 | 1 | 47 |
| 2 | **1972** | **2** | **1** | **4** | **21** | **26** |
| 3 | 1974 | 1 | 4 | 17 | 39 | 22 |
| 4 | **1958** | **25** | **13** | **46** | **5** | **49** |
| 5 | 1956 | 10 | 7 | 1 | 11 | 50 |
| 6 | 2002 | 15 | 19 | 26 | 8 | 23 |
| 7 | 1967 | 7 | 9 | 28 | 7 | 39 |
| 8 | **1999** | **4** | **8** | **44** | **45** | **1** |
| 9 | 1997 | 5 | 2 | 9 | 22 | 13 |
| 10 | 1964 | 18 | 16 | 33 | 26 | 15 |

# Figures

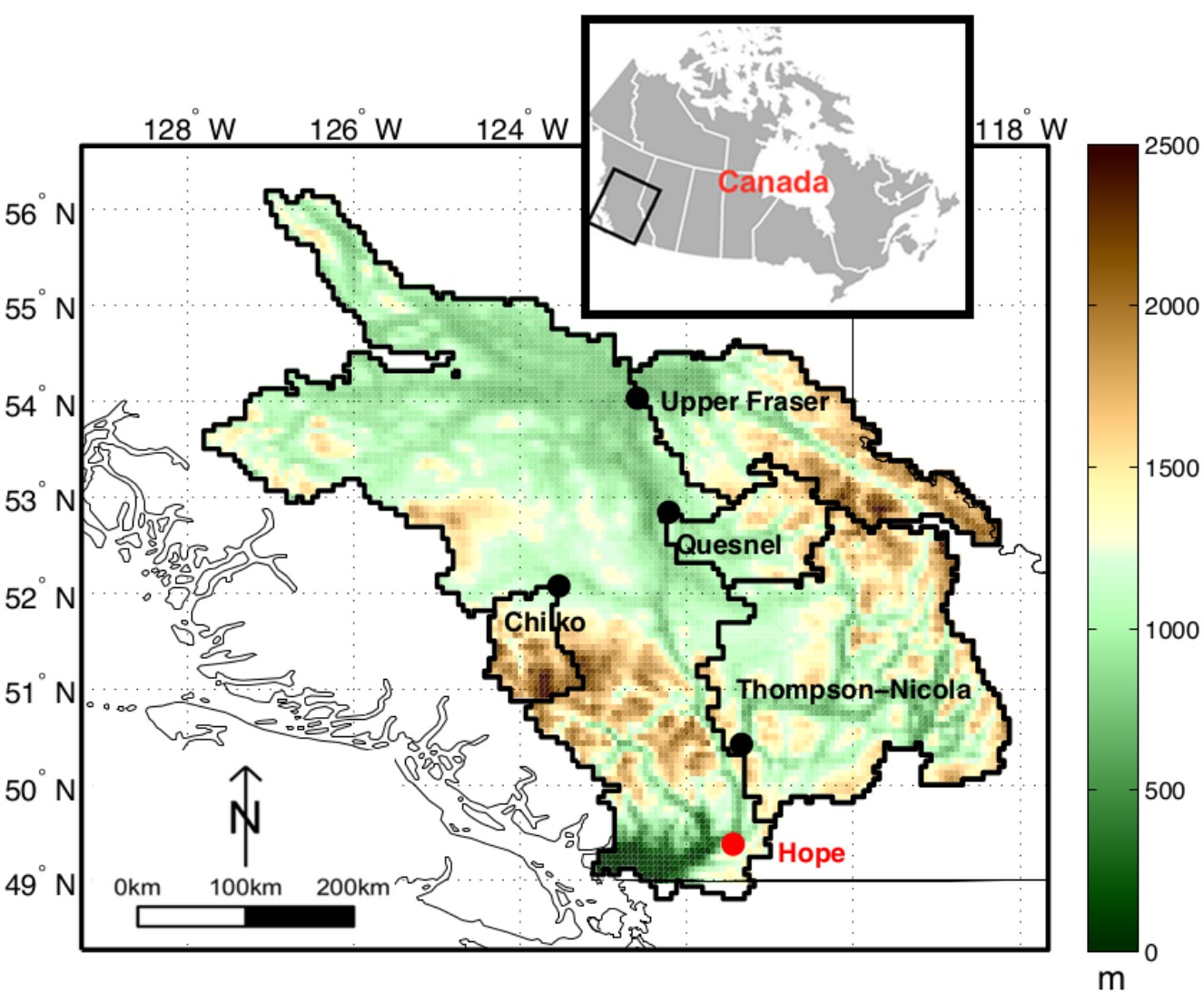

Figure 1: The Fraser River Basin of British Columbia, Canada and four of its subbasins examined in this study. Colours correspond to elevations as used by the VIC model, at 1/16° horizontal resolution. Locations of streamflow gauge stations at the outlet to each subbasin are shown with black dots, while the major outlet for the FRB at Hope is shown with a red dot. Details of each subbasin are provided in Table 1.

**Figure 2: a) Regression of observed annual peak daily streamflow (APF) magnitude at Fraser-Hope hydrometric station (vertical axis) against the NINO3.4 index over the period 1912-2014. Year labels are plotted as individual points. b) Same as in a) but for the PDO index. Years in boldface font are discussed in the text of Sec. 3.2. c) Percentiles of APF in years when the NINO3.4 index is in the negative (La Niña) phase (filled circles) versus the positive (El Niño) phase (open circles). The solid curve shows the median result of resampling the combined data set of both ENSO phases $10^4$ times. The significance of differences between APFs in either phase and corresponding samples of the combined data set is assessed using a permutation test (Sec. 2.3), resulting in the indicated *p*-values. d) Same as in c) but for the PDO index.**

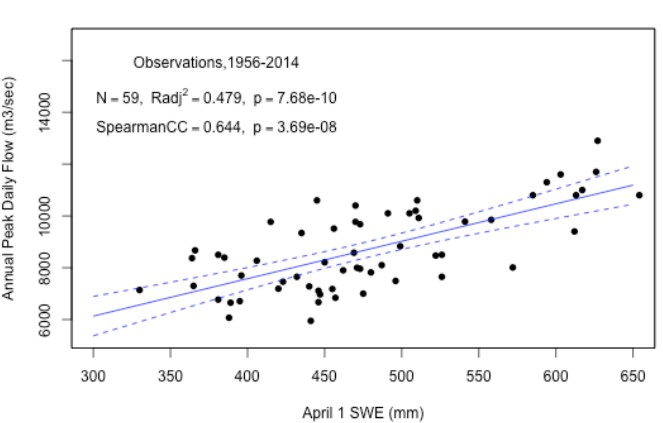

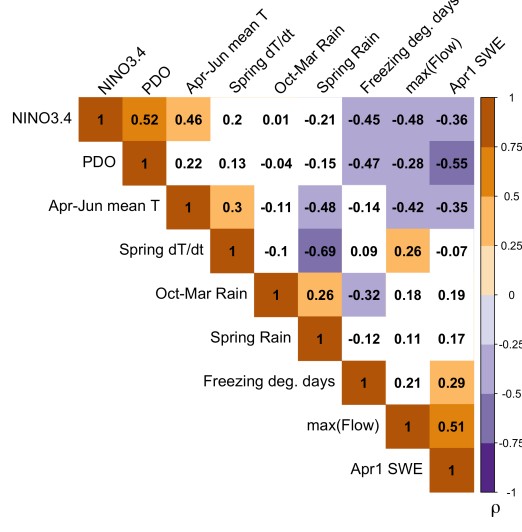

**a**

**b**

**Figure 3: a) Regression of observed APF against basin-averaged 1 April SWE over the Fraser Basin, within the period of overlapping records, 1956-2014. The solid blue line is the least squares linear regression fit, while the dashed curves show the 95% confidence interval. Adjusted Pearson and Spearman rank correlation coefficients, along with their associated *p*-values, are indicated at upper left. b) Correlogram of all observed variables, averaged over the Fraser Basin over the common period of records for all variables, 1956-2006. Numerical values are Spearman rank correlation coefficients, and coloured squares indicate significant correlations at the *p* < 0.1 level.**

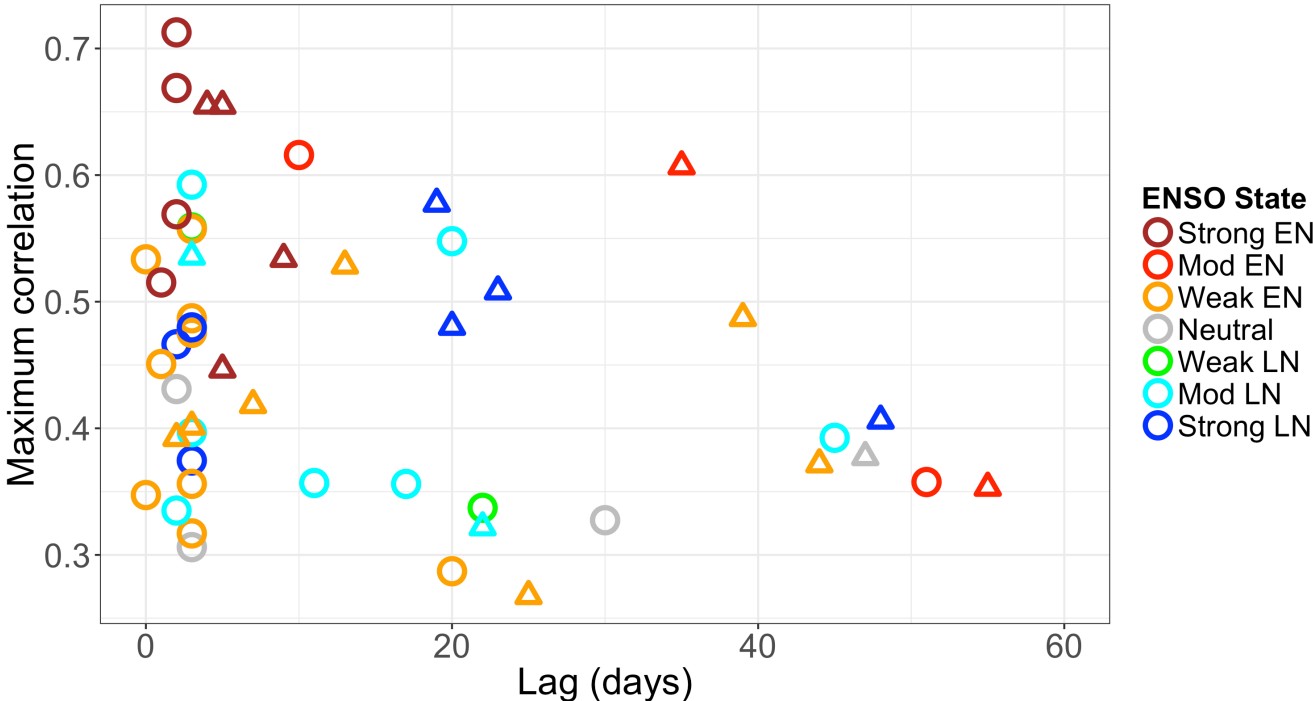

**Figure 4: Results of lagged regression of FRB-averaged daily rainfall, characterized by the current rainfall index (CRI), on streamflow at Fraser-Hope. Only the subset of years wherein there exists a significant relationship between CRI and discharge are shown. The ENSO state, as specified in the legend according to colour (EN: El Niño, LN: La Niña), corresponds to the year preceding the rainfall and discharge, and was obtained from http://ggweather.com/enso/oni.htm, based on the same definition used in the analysis of Sec. 3 (i.e. NINO3.4). The maximum correlation, max($\hat{\rho}_{XY}$), is shown on the vertical axis, with the corresponding lag, $\tau$, between rainfall and discharge on the horizontal axis. A positive lag means that streamflow lags CRI. *Circles*: Observed precipitation versus observed discharge over the 1950-2006 period. *Triangles*: Observed precipitation versus VIC-simulated discharge. All correlations are significant at the $p < 0.05$ level. Two outlier points with lags > 60 days (with none exceeding max($\hat{\rho}_{XY}$) = 0.42) are omitted from the plot.**

**a**

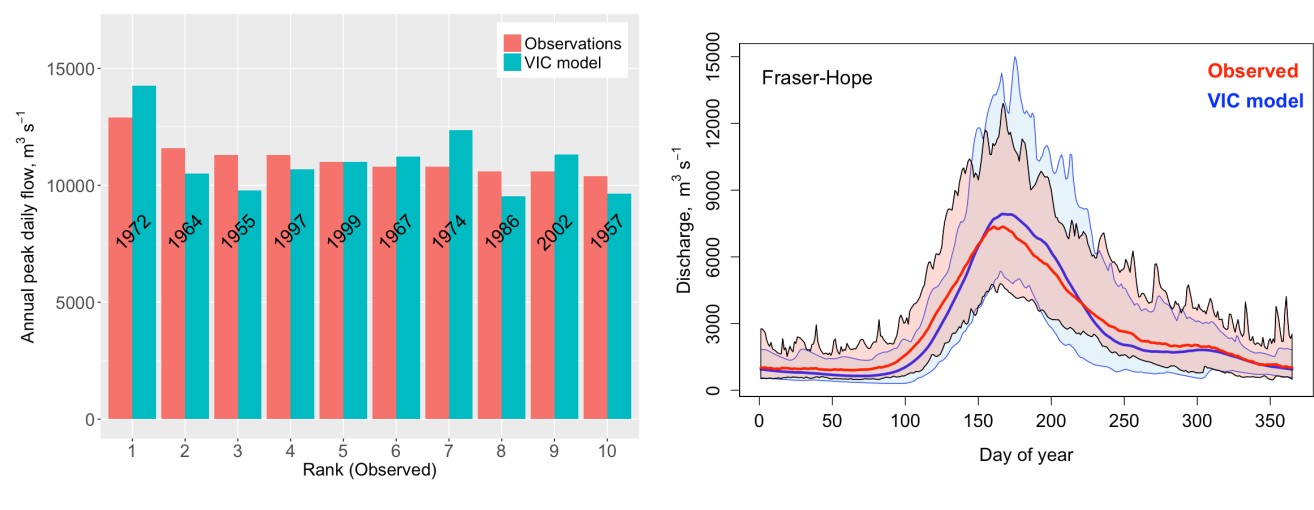

**b**

**c**

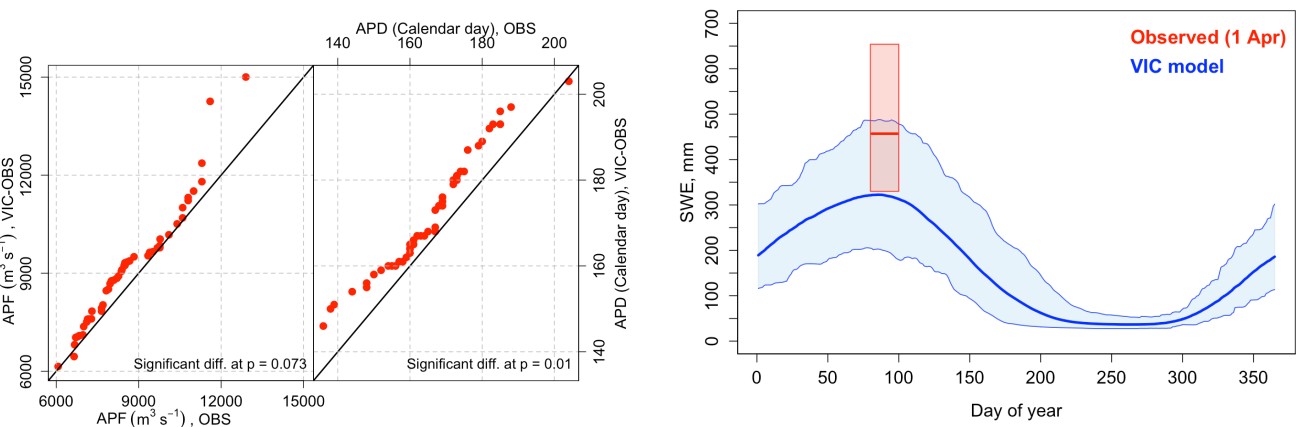

**d**

**Figure 5: a) Annual peak daily streamflow (APF) magnitude in observations (red) and in the VIC simulation (blue) at Fraser-Hope, ranked by observed values over 1955-2004. b) Annual cycle of daily streamflow in observations (red) and VIC simulation (blue) over 1955-2004 at Fraser-Hope. c) Quantile plots of VIC-simulated versus observed APF and APD at Fraser-Hope. Differences between the respective distributions are assessed using a permutation test with $10^4$ resamples; the resulting *p*-values are indicated in each sub-panel. d) 1 April observed SWE (red) compared with annual cycle of SWE from VIC, both averaged over the FRB for 1956-2006. Heavy curves in panels b and d show the multi-year mean while shaded areas show the interannual range.**

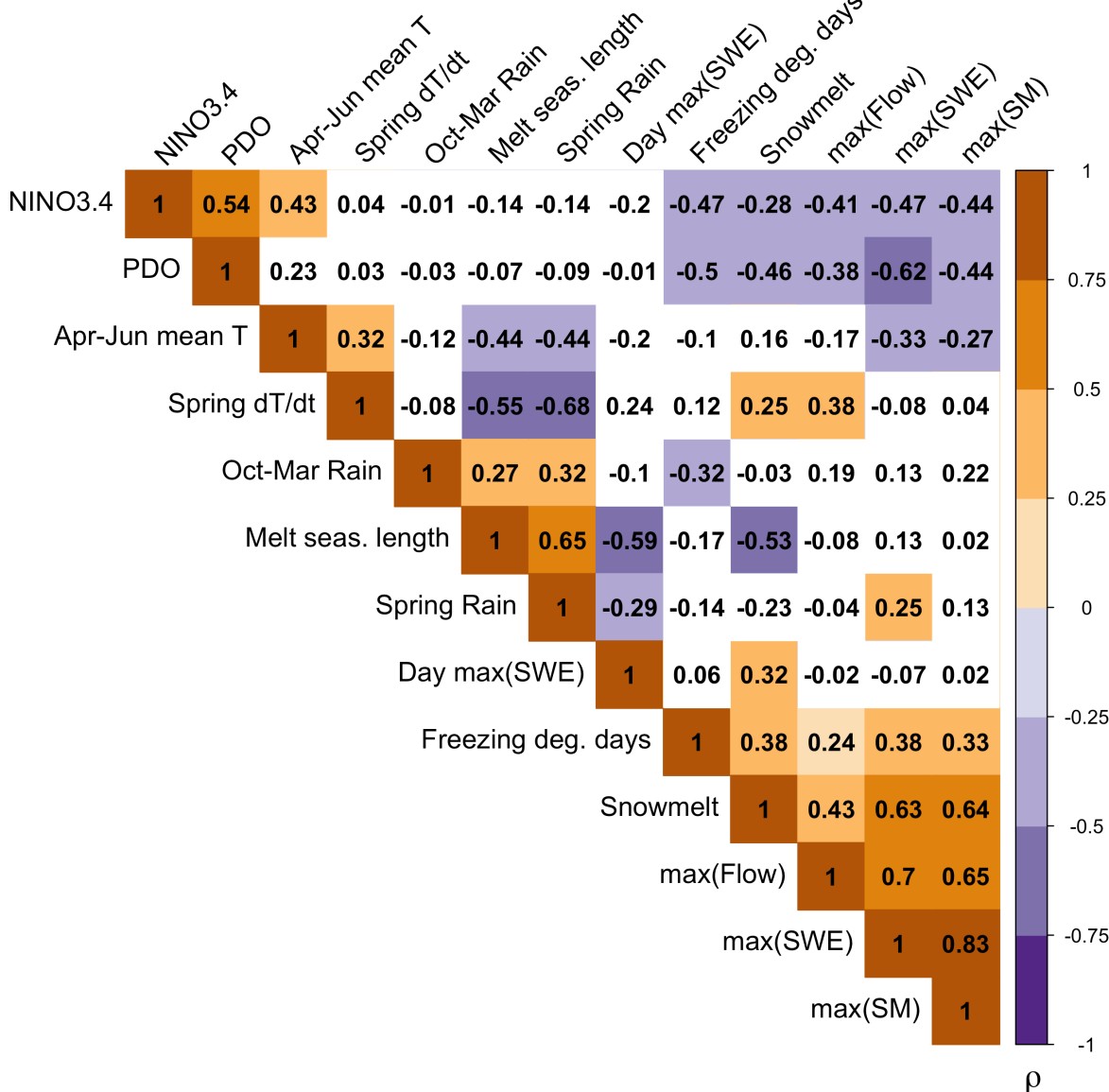

**Figure 6: Correlogram of all observed and VIC-simulated variables, averaged over the Fraser Basin within the common period of records, 1955-2004. All values are Spearman rank correlation coefficients, and coloured squares indicate significant correlations at the $p < 0.1$ level. A hierarchical clustering algorithm has been applied to group like-with-like correlations.**

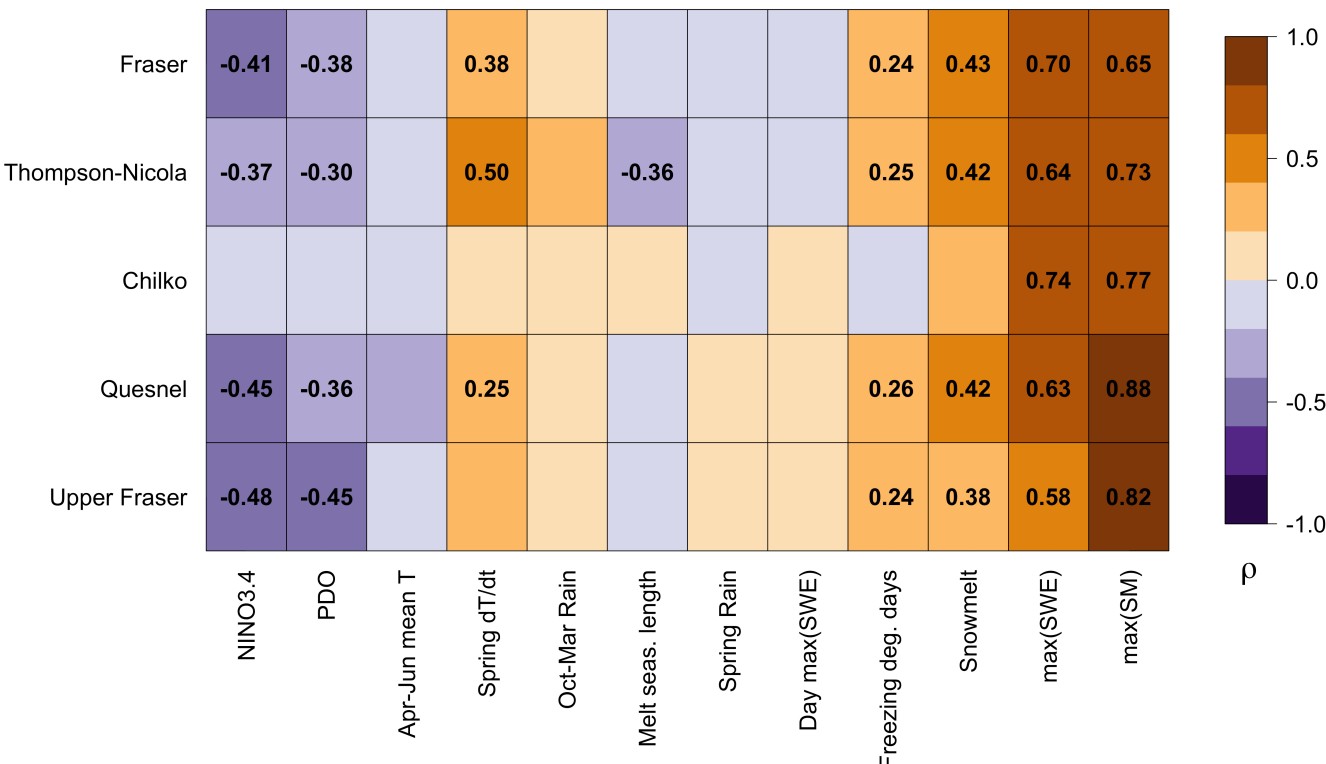

**Figure 7: Correlogram of observed and VIC-simulated variables, averaged over each of the indicated subbasins within the 1955-2004 period, against APF at the subbasin outlet (Table 1). The cell values and colour scale indicate Spearman rank correlations, significant at the $p < 0.1$ level. The ordering of variable columns is the same as in Fig. 6, for ease of comparison.**

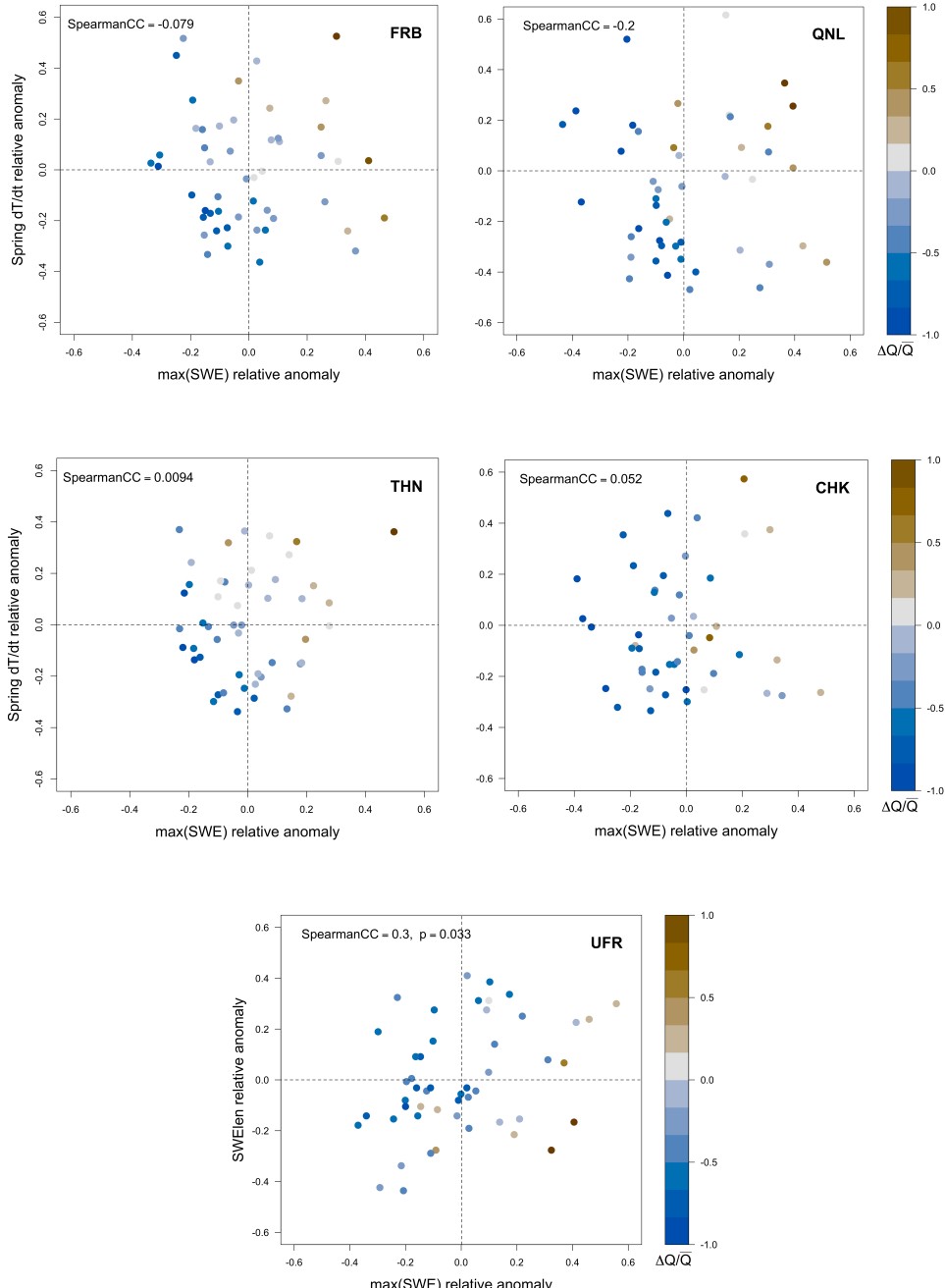

**Figure 8: Scatterplots showing the co-dependence of relative APF, $\Delta Q / \overline{Q}$, (colour scale) on the two principal predictors from the MLR analysis (*x*- and *y*-axes), expressed as relative anomalies, in each basin. Counterclockwise from top left: entire Fraser (FRB), Thompson-Nicola (THN), Upper Fraser (UFR), Chilko (CHK) and Quesnel (QNL). The primary predictor in all subbasins is SWE$_{max}$, while the secondary predictor is *dT/dt* in all subbasins except UFR, where it is SWE$_{len}$. The Spearman rank correlation between the two predictors is shown at upper left in each panel; other than in UFR, none are significant at the 10% ($p < 0.1$) level.**

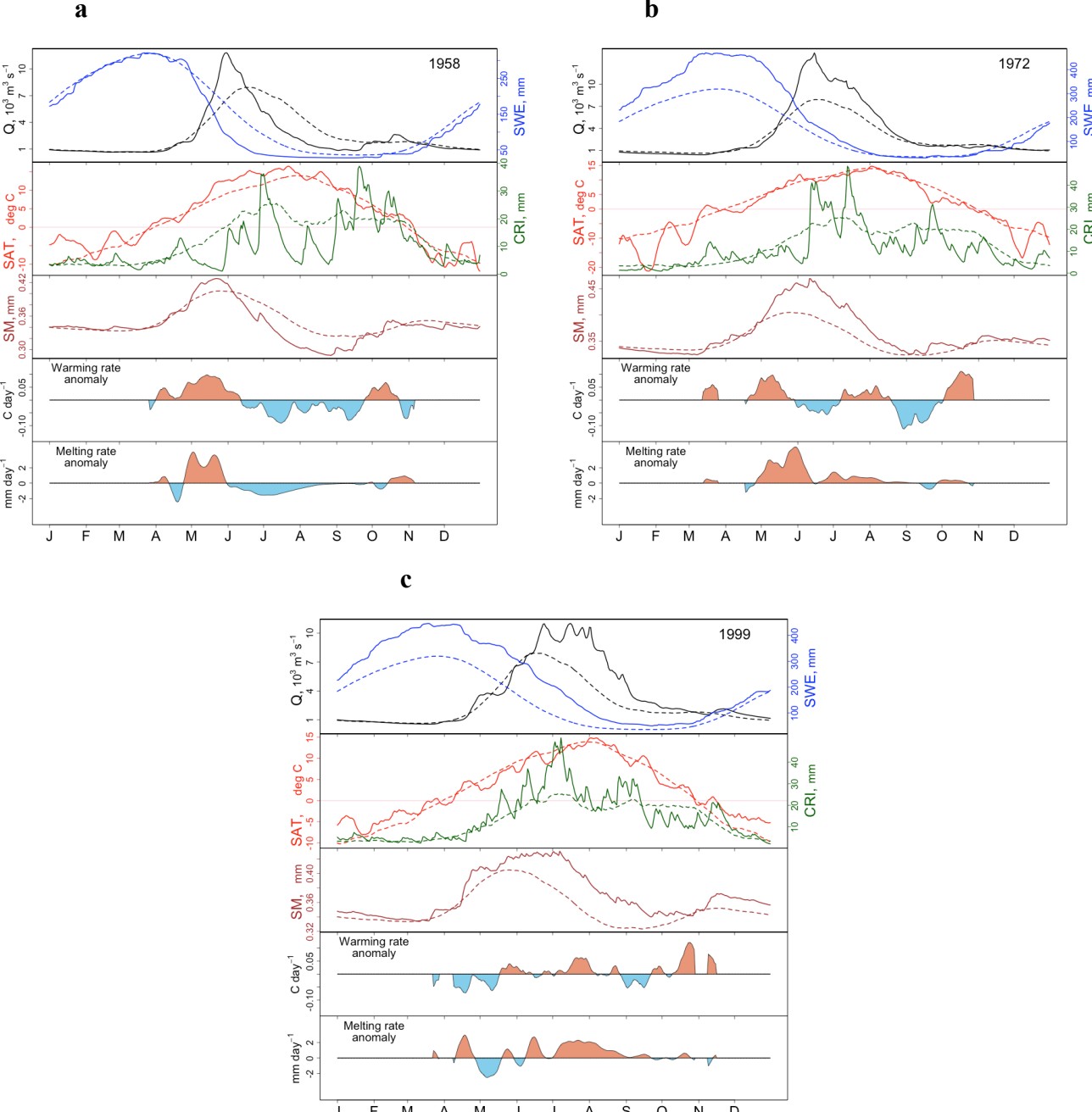

**Figure 9: Annual cycles of key variables from the VIC simulation, constructed from daily output. Results for three high-APF years are shown: a) 1958; b) 1972; and c) 1999. In each panel, sub-panels show, from top to bottom: streamflow (black, left-hand axis); SWE (blue, right-hand axis); surface air temperature (red, left-hand axis); current rainfall index, CRI (green, right-hand axis); soil moisture (brown); warming rate anomaly; and melting rate anomaly. In the upper three sub-panels, solid curves show time series for the year indicated at top right, while dashed curves show the multi-year (1955-2004) mean climatology. The calculations underlying the lower two sub-panels are described in Sec. 4.**