# Peer review of "Examining controls on peak annual streamflow and floods in the Fraser River Basin of British Columbia"

_Hydrology and Earth System Sciences, 2017_

## Referee Comment (RC1) · Anonymous Referee #1 · 25 Sep 2017

Overall, this is a careful and interesting statistical study of what controls the magnitude of streamflow in a particular large river basin. While there doesn't seem to be anything technically wrong with the analysis, I feel that the paper can be improved significantly by considering the following points:

Major comments:

1. Given the emphasis on streamflow "predictors" and the explicitly indicated application of seasonal streamflow prediction (e.g., in the abstract), it seems to me that the study would be much more useful if different sets of statistics were computed for different prediction lead times. For example, on the face of it, it doesn't make much sense

to consider an SST index based on a previous season's SST conditions (p. 6, lines 10-14) along with a warming rate (p. 10, line 16) that apparently can only be computed on the day of the APF itself; if one were to wait so long into the high streamflow season to estimate APF from various variables, mightn't they use concurrent SSTs, which may have changed significantly? Another example: the multiple regression (equation 1) connects SWE on April 1 with a heating rate (dT/dt) that uses information up to the date of the APF. Someone interested in predicting subsequent APF given May 1 conditions presumably would want to use May 1 SWE and the heating rate in the period leading up to May 1 as the predictors (along with the average multi-month temperature up to May 1). Multiple correlograms could in fact be constructed, based on a selection of different forecast start dates (i.e., based solely on information available on those start dates). Such an expanded set would be more informative and intuitively more understandable than the single set provided here, given that the single set draws on information at such different leads.

Perhaps the main usefulness of the study is not to predict APF but (as also mentioned in the abstract) to project future APF behavior based on projected changes in the different predictors. That may be, but then some discussion is presumably needed about how well one can possibly project such changes in the predictors. We have a hard enough time convincing people that we can predict precipitation changes at the basin scale, let alone the more specific predictors discussed here.

2. The authors note the potential for soil moisture state to influence streamflow but don't include it in their calculations for the observational analysis (section 3.3) due to the unavailability of data. Why not use, as a surrogate, the average fall rainfall or summer+fall rainfall prior to the (estimated) time of soil freezing?

3. The case studies are fine, as far as they go. The authors should remember, however, that the typical reader is not specifically interested in the Fraser River Basin; they are reading the paper for a more general understanding of what controls streamflow magnitudes. In my opinion, the case studies don't add all that much; they could be

removed to make room for the expanded analyses suggested in Comment 1 above.

Minor comments:

– Table 3: Change "APDF" to "APF".

– p. 11, line 9: How can APF be said to depend on R_APF, when the latter includes information subsequent to the date of APF? A confusing variable to consider (though I don't think I see it in the correlogram, anyway). . .

– The point of Figure 4 is not as intuitive as the authors assume. More description is needed to describe the whole CRI thing.

– p. 15, lines 16-17: How does the recharging of soil moisture increase further melting? This doesn't make sense.

---

## Referee Comment (RC2) · M. Jenicek (Referee) · 6 Oct 2017

General comment

The objective of presented study is to define and analyse major factors which influence annual peak flow (APF) in the Fraser River basin (British Columbia). I think authors did an interesting work by analysing the statistical importance of several snow, meteorological and soil signatures, which influence the occurrence and extremity of APFs in the Fraser River and its tributaries. Besides observations, they used a hydrological model, which enabled to simulate various components of the water balance.

In my opinion, the results are interesting, although not surprising as they mostly support our existing qualitative knowledge of how different natural drivers influence catchment runoff. Although, I did not find many new findings in the paper, I am convinced that the quantification of the role of different drivers on runoff is a valuable and novel contribution. Therefore, I found results of this study important and certainly appropriate for HESS. However, I have some comments listed below, which need to be addressed before I can recommend the manuscript for publication.

Major comments

Although, the introduction section is nicely written, I am missing some larger context coming out from presented studies. Authors were mostly concentrated on the Fraser River basin, but there would be certainly interesting to provide the reader also with larger context by focusing on other regions of the world with similar climates and runoff regimes.

Section 2.3 and Table 2: Authors used three signatures related to rainfall (cold season rainfall, spring rainfall and APF rainfall). However, none from these three signatures reflects the decreasing importance of rainfall amount in time, which might be important when relating these rainfall amounts to APF. Why authors did not use e.g. Antecedent precipitation index API or Current rainfall index CRI (calculated e.g. for the day of year with APF)? Besides, I am a bit confused because authors presented CRI in the result section 3.3.3 and Fig 4, but this index was not included in correlation matrixes (Fig. 3, 6 and 7).

Page 8, lines 22-25: One of the assumption when using multilinear regression (MLR) is that the individual predictors should be mutually independent. How authors dealt with this limitation? I found several times in the text that all possible combinations of predictors listed in Table 2 have been used, but it seems that some of them might be mutually dependent (e.g. spring warming rate and snowmelt rate or NINO3.4 and PDO indexes). Perhaps, the mutual dependence might be also the reason why PDO and

[Figure]

NINO indexes were not selected by the MLR as statistically important.

Sections 3.3.5 and 3.5.5: I am not sure whether the results from the MLR could be somehow practically used. Authors introduced an equation to calculate APF based on selected predictors, but obviously some of these predictors are not known before the start of calculation. Specifically, dT/dt is not known before APF occurs, so it is not much useful to use it as a predictor. Authors are aware of it by mentioning this issue on page 22, lines 9-19. However, I would expect more discussion on this topic. Maybe, authors should not concentrate much on APF prediction in some specific year, but they could rather discuss how future changes in predictors (e.g. future increase in air temperature) might influence the APF in the future. I am aware that such discussion might be a bit of speculative manner (since the future changes in predictors was not the scope of the paper), but it could still bring some new insight on results.

Overall, I am a bit uncomfortable with the paper structure. Authors sometimes mix methods, results and discussion together. Some methods are described in the results section (e.g. current rainfall index in section 3.3.3; dealing with "glacier cells" in section 3.4). Similarly, there are a lot of discussion text in the results section. Therefore, "Discussion and conclusions" section is just a repetition of major results without any discussion. I suggest to better separate the individual parts. In my opinion, authors could either make a separate section called "Discussion" and put all related text into it, or (which might be an easier solution) they could make a section "Results and discussion" where they can put all related information. I do believe that this way the text will be clearer to the reader.

Minor comments and technical corrections

The abstract sounds a bit vague to me. Maybe, authors could present their results in more detail (some numbers etc.).

The aims of the paper are not explicitly defined in the introduction section (although, they are intuitively deductible from given context).

Page 3, line 5; page 15 line 28: I think "On the other hand" is not correctly used here since there is no "on the one hand" before.

Page 4, lines 11-13: I fully agree with this statement. However, I did not find any section where authors discuss this issue within the context of their results (maybe, I just missed it).

Page 6, line 4: Undefined abbreviation (WSC).

Page 6, lines 8-10: I think there should be more explanation about the mentioned 19 locations. Were they equally distributed in the basin area? Do they cover whole elevation range? This should be shortly mentioned in the text.

Page 6, line 12: The SST is defined only later (line 15). This abbreviation should be defined already here.

Page 6, line 18: Missing year in reference Gurrapu et al.

Section 2.2: For those, how are not familiar with VIC model, I would expect at least a brief introduction of model routines (specifically, snow and soil routines since they seem to be most important for analyses presented later in the paper). It might help readers with understanding such they do not need to find information about the model in other literature.

Page 7, lines 10-17: The calibration and validation procedures should be described. Which objective function(s) has been used? Was only one "best" parameter set used for further simulations or more parameter sets were used to decrease the uncertainty resulted from calibration? This should be clarified. Additionally, it was not fully clear to me, whether authors did the model calibration by themselves or they used simulations done by someone else. It seems that latter is true, but it is not fully clear from the text.

Page 7, line 8: What exactly means "computational constrains"? Please, specify.

Page 7, line 14: "...were in better agreement with observation...". Based on which

criteria?

Page 11, lines 17-20: The procedure describing how to compute CRI should be placed rather in methods section.

Page 13, lines 27-33: Why model simulated mentioned unrealistic SWE in high-elevation grids? I am not familiar with VIC model, but unrealistic increase in SWE at highest elevations (sometimes called as "snow towers") exists also in some other hydrological models and this problem is usually connected with simplified degree-day method used in snow routine. Typically, when 1) snow routine uses only one value of melt factor over the whole melt season and thus it underestimates the radiation inputs and/or 2) snow routine does not account for snow redistribution due to wind.

Page 14, lines 1-5: I think this paragraph should be placed in methods.

Page 15, lines 8-9: Could dT/dt explain the mentioned large APFs occurred during average snow conditions?

Page 15, lines 22 and 23: missing values of p.

Page 18, line 6: Maybe, I would not title this chapter as "Case study" since basically everything presented in this paper is a case study. Thus, I suggest to simply title it "3.8 High-flow years in the FRB".

Table 2: I would prefer "Day of year (DOY)" instead of "Julian day" since the Julian day is the continuous count of days since the beginning of the Julian period.

Table 3: I guess there should be "APF" instead of "APDF".

Figure 1: Missing scale. Additional, the colors in the color scale are not appropriate for DTM (red to blue is rather used for air temperature).

Figures 2, 3, 5, 8 and 9: Maybe, the size of captions (axis, legend) are too small.
* * *
[Figure]

531, 2017.

---

## Referee Comment (RC3) · S. Dery (Referee) · 15 Oct 2017

Summary: This is a very interesting effort highly suitable for publication in Hydrology and Earth System Sciences. The paper is very well-written and figures are generally clear and entirely appropriate to illustrate key points. Perhaps one weakness is that the structure of the paper is confusing, with a mixture of methods, results and discussion within the Results section. This report provides guidance that the authors should consider in revising their manuscript.

General Comments:

[Figure]

1) The structure of the manuscript requires some reorganization, as methodological information is at times provided in the Results section. All aspects of the methods should be provided in Section 2, which should improve the legibility of the paper. Similarly, the Discussion is rather brief and needs to be enhanced to better describe the implications of the results and potential applicability to other snow-dominated watersheds worldwide.

2) Apart from manual snow surveys (usually conducted on a bi-weekly basis), the British Columbia (BC) River Forecast Centre operates an extensive network of snow pillow stations that provide daily snow depth, snow water equivalent (SWE) and meteorological data in real-time. Why not incorporate the data from the snow pillow network within and near the study area (FRB) to improve the potential predictability of annual peak flows in the Fraser River? The snow pillow data are an integral part of operational flood forecasting in BC and so incorporating those data in the statistical models would inform provincial hydrologists on the importance of those snow data in predicting annual peak flows (APFs) in the Fraser River.

3) Why are some of the larger subbasins of the FRB not included in the present study? The Nechako, Blackwater (West Road) and Chilcotin Rivers all form important tributaries to the Fraser River with generally complete observational records from the early 1950s onward (early 1970s for the Chilcotin River) and could provide further regional insights on the contributions of these systems to the APFs observed on the Fraser River at Hope.

4) Overall, which subbasins contribute most to the observed APFs on the Fraser River at Hope and what are the lag times between their high flows and those downstream? Presumably information on upstream flows alone can provide a measure of predictability on the potential high flows for the Fraser River at Hope. . .

Specific Comments:

1) P. 2, lines 3-14: The introduction could highlight the ecological importance of the

Fraser River, namely its globally-significant salmon populations who migrate twice through the river to/from the Pacific Ocean during their lifetimes.

2) P. 2, line 10: The abbreviation for British Columbia was defined on line 3, so use it henceforth. A similar issue arises with other abbreviations that are not necessarily used after being defined.

3) P. 2, lines 19-20: Why provide here the details of the Water Survey of Canada hydrometric gauge on the Fraser River at Hope, BC? This detailed information can be provided in the Data and Methods section.

4) P. 3, lines 16-18: Note that Déry et al. (2012) explored trends in the variability of flows at 139 sites across the FRB while Hernández-Henríquez et al. (2017) assess trends in flows for 136 sites for rivers draining BC's Coast and Insular Mountains. While there is some overlap between the two datasets used in these efforts, they remain two independent studies with different goals, study areas and streamflow datasets.

5) P. 4, line 23: Rather than 'private communication', this should likely be 'personal communication'.

6) P. 5, line 4 and elsewhere: Date formats may need to be changed depending on the journal's preference, i.e. perhaps '5 June'.

7) P. 5, line 6 and elsewhere: Use superscripts for all units, i.e. 'm3 s-1'.

8) P. 6, line 3: Here again abbreviations for the FRB and BC could be used.

9) P. 6, line 18: Add the year of this publication.

10) P. 6, lines 22 and 23: Spell out "WNA" and "PRISM".

11) P. 7, line 3: How many elevation bands are used in the VIC simulations?

12) P. 7, lines 19-24: Moore and McKendry (1996) and Hsieh and Tang (2001) both provide relevant information on the impacts of large-scale teleconnections on BC's

snowpacks and should therefore be included as important references here. Whitfield et al. (2010) also provide a comprehensive overview of the impacts of the Pacific Decadal Oscillation on the hydroclimate of western Canada.

13) P. 7, line 31: Fix comma before "see".

14) P. 8, line 28: Why are trends inferred from linear regressions and not the Mann-Kendall test instead?

15) P. 9, line 9: Why focus only on 1 April SWE when many sites across the FRB experience their peak annual accumulations near 1 May (e.g., Déry et al., 2014)?

16) P. 11, line 22: Should this be "latter half"?

17) P. 12, lines 28-30: What are the units for the variables in Equations (1) and (2)?

18) P. 15, line 23: The 'p-value' is missing here.

19) P. 17, line 7: Here again units are missing for the variables in the multivariate regression (Equation (3)).

20) P. 21, line 13: Delete 'in order'.

21) P. 22, line 5: Same comment.

22) P. 22, line 30: Please provide the full range of pages for this reference.

23) P. 23, line 29: Add the article number for this reference.

24) P. 24, line 2: Add the full range of pages for this reference.

25) P. 24, line 19: Add the volume and correct page range for this reference.

26) P. 25, line 18: Add the article number for this reference.

27) P. 27, line 20: Update this reference if this paper has now been published in the regular section of HESS.

28) P. 28, Table 1: Why are the Upper Fraser, Quesnel, Thompson and Chilko Rivers selected as subbasins for this study? Why does the period of record end in 2013 for three of these rivers?

29) P. 36, Figure 5(a): Are the discrepancies between observed and VIC-simulated flows for the APFs due to the lack of a representation of the interbasin diversion of flows from the Nechako Reservoir and River towards the Kemano River?

References:

Déry, S. J., Hernández-Henríquez, M. A., Owens, P. N., Parkes, M. W., and Petticrew, E. L.: A century of hydrological variability and trends in the Fraser River Basin, Env. Res. Lett., 7, 024019, 2012.

Déry, S. J., Knudsvig, H. K., Hernández-Henríquez, M. A. and Coxson, D. S.: Net snowpack accumulation and ablation characteristics in the Inland Temperate Rainforest of the Upper Fraser Basin, Canada, Hydrology, 1, 1-19, 2014.

Hernández-Henríquez, M. A., Sharma, A. R., and Déry, S. J., 2017: Variability and trends in runoff in the rivers of British Columbia's Coast and Insular Mountains, Hydrol. Proc., 31(18), 3269-3282.

Hsieh, W. W. and Tang, B.: Interannual variability of accumulated snow in the Columbia basin, British Columbia, Water Resour. Res., 37, 1753-1759, 2001.

Moore, R. D. and McKendry, I. G.: Spring snowpack anomaly patterns and winter climatic variability, British Columbia, Canada, Water Resour. Res., 32, 623-632, 1996.

Whitfield, P. H., Moore, R. D., Fleming, S. W., and Zawadzski, A.: Pacific Decadal Oscillation and the hydroclimatology of western Canada – Review and Prospects, Can. Water Resour. J., 35, 1-27, 2010.

---

## Author Response (AR1)

**Authors' response to Reviewer and Editor comments**

We thank the Reviewers and Editor for their perceptive and helpful comments. Our point-by-point response to the comments is given below, organized by reviewer.

*Response to Reviewer 1 comments:*

1) We should clarify that our work is not focused upon predictability of APFs in an operational sense, i.e. flood forecasting on the annual or seasonal time scale. Our objective is instead to discover and quantify relationships between the interannual variability of APF, large-scale climate indices, and various basin-averaged variables within the FRB. We regret that the last sentence of the Abstract led the Reviewer to believe that the former was the primary motivation. Hence, we have modified this statement to clearly distinguish between what we did ("identification of these controls") from possible applications of the work ("understanding seasonal predictions or..."). Moreover, only the former is emphasized earlier in the Abstract (lines 18-22).

As for our use of the word "predictor", this term has a clear, if broadly applied, meaning within the context of physical and statistical models. It simply denotes a quantity that covaries with some other variable of interest, the predictand. Moreover, it retains this meaning in either a retrospective (past) or prospective (future) sense. The usage of "predictor" and "predictand" thus parallels that of "independent" and "dependent" variables, but is better suited to the statistical model used here to uncover the relationships between physical variables. Hence, we feel that our use of the term "predictor" is well-suited to the methods of the study. Nevertheless, we have inserted sentences at a number of locations in the text to clarify our use of these terms.

Moving to the remainder of the Reviewer's comment, we agree that the predictors we chose might not be suitable for the prediction of APF in an operational context. But as stated above, we are interested instead in a more retrospective approach that accepts that any predictor, at any lag, might have explanatory value for the predictand. Indeed, there is a physical reason why a NINO3.4 index based on concurrent SSTs would not be expected to be an effective predictor of APF: namely, because teleconnections between tropical SST anomalies and the climate of the FRB operate with a lag of ~2-6 months (Gurrapu et al. 2016). So even in an operational context, using a lagged SST index as a predictor makes sense.

The potential use of our results in the context of future projections was also a question raised by Reviewer 2, which we address in that portion of the Response below.

2) We thank the Reviewer for this suggestion. Using the observed data, we regressed both autumn (Sept-Nov) and summer-autumn (Jun-Nov) rainfall on the following year's APF over the 1950-2006 period. Neither correlation was significant at the 10% significance level.

3) We appreciate the Reviewer's opinion on this matter, and see considerable overlap between it and the first major comment of Reviewer 2: see that Response, which details the significant revisions made. We feel that the three case studies exhibited in Sec. 4 effectively illustrate phenomena highlighted in the preceding regression analyses and that, consequently, are likely to extend beyond the specific context of the FRB. This is encapsulated in the subsection titles, which point to the climate drivers underlying the different hydrographs seen in the sampled years. Finally, as we have argued against expanding the analysis in connection with point 1) above, we feel that there is no pressing need to make cuts to existing material elsewhere in the manuscript.

*Response to minor comments of Reviewer 1:*

- Changed "APDF" to "APF" in Table 3

- p.11, line 9: We thank the referee for this question, which makes a valid point. We have redefined R_APF to span only the 15-day period immediately prior to the date of APF, with no change in the conclusions.

- Figure 4 and the CRI: The CRI is introduced in an attempt to track short timescale influences of multi-day rainfall on daily streamflow. These intra-annual influences, if present, will not be detected by the interannual regressions described earlier in Sec. 3, which involve longer-term means of R versus annual peak flow only. This motivation is now clearly explained in the third paragraph of Sec. 3.3.3. Fig. 4 summarizes these intra-annual covariations, as opposed to the interannual relationships examined thus far in the paper. Fig. 4 also identifies the ENSO state corresponding to the year prior to that of each (rainfall, discharge) data point on the graph, consistent with the convention defined in Sec. 2.1.

- p.15, line 16-17: We agree that this phrase is unsubstantiated. It has been replaced with a clearer sentence reinforcing the relationships found via regression.

*Response to Reviewer 2 comments:*

1) We are grateful for the Reviewer's comment on the Introduction, which highlighted two weaknesses in the submitted version: first, the larger context and motivation for the work were incompletely expressed; and second, the historical summary provided in Sec. 1.2 is something of a distraction for those readers not specifically interested in the FRB. For this reason, we have eliminated much of the material from the previous Sec. 1.2 and moved two shortened paragraphs to Sec. 4, where they fit well with the case studies examined there. A new Sec. 1.1 better explains the context and aims of our work.

2) Section 2.3 and Table 2: As mentioned in the response to the minor comment of Reviewer 1 above, we have provided a clearer motivation for our use of the CRI in the last paragraph of Sec. 2.3 and in the paragraph describing the cross-correlations between CRI and streamflow in Sec. 3.3.3. The measure R_APF is similar to CRI, but weights each day's rainfall equally in the sum, while CRI weights each prior day's R by an additional factor of 0.9. Thus, as mentioned by the Reviewer, R_APF likely overestimates the influence of antecendent rainfall in the regression calculation compared with CRI. Since we found that R_APF was not a significant predictor of either observed or VIC-simulated APF, we might expect that CRI computed for the day of APF would be a similarly poor predictor of APF. An explicit check confirms this conclusion. Hence, it is something of a moot point whether R_APF or CRI(day of APF) should be included in the predictor set, as neither is significantly correlated with APF (and thus neither appears in the correlograms).

Despite these negative results for an interannual relationship between CRI and APF, we emphasize that intra-annual covariations between CRI and streamflow at various lags are detected, as summarized in Fig. 4.

3) p.8, lines 22-25: MLR does not require the "mutual independence" of predictors in the sense of the predictors being orthogonal, but it does require that the predictors are not multi-collinear. That is, if there are *k* predictors, the predictors should collectively span a *k*-dimensional vector space as opposed to a lower-dimensional vector space. Ascribing "explained variance" to individual predictors when predictors are not orthogonal is always challenging. One approach that is sometimes used is to orthogonalize the predictors, but this produces explanatory variables that are difficult to interpret, are not directly observed and in fact, cannot be determined

uniquely (any fixed rotation or reflection of a given set of orthogonal vectors would produce another set of orthogonal vectors that spans the same vector space equally well). We therefore prefer to use the candidate predictors as observed, recognizing that they are not fully independent and taking that into account in the interpretation of the MLR results. The extent to which the predictors are linearly dependent on each other can, in part, be assessed by calculating cross-correlations, which are discussed in Sec. 3.3.4 and illustrated in the correlograms Figs. 3b and 6.

4) Sections 3.3.5 and 3.5.5: The Reviewer correctly ascertained that the MLR models we obtain are not suitable for APF forecasting. Instead, the word "predictor" is used in a statistical sense; "predictor" is often used as a synonym for "independent variable" when regression models discussed. Rather, they constitute a useful summary of which variables have most influenced the APF over the historical period examined. The Reviewer (and also Reviewer 1) also asked about whether results from the MLR might be used to say something about projected future APFs. We have made some suggestions along these lines in the Conclusions.

5) Paper structure: In response to this criticism, and acknowledging similar comments from Reviewer 3, we have reorganized the content in the revised manuscript. Specifically, the descriptions of methods in the Results were moved to Section 2: Data and Methods.

*Response to minor comments of Reviewer 2:*

"…abstract sounds a bit vague…(some numbers, etc.).": We have added the univariate correlation coefficient value for each predictor mentioned in the Abstract, in order to provide a more detailed description of results.

"…aims of the paper are not explicitly defined in the introduction…": As mentioned in the response to point 1) above, a new Sec. 1.1 better explains the context and aims of our work.

Page 3, line 5; page 15 line 28: In fact, there is no standard in English usage requiring that "On the other hand," be preceded by "On the one hand". It is acceptable to use this phrase when it is clear what the contrasting situation is that is being referred to, even if that contrasting situation or thought is implicit. Specifically, the former phrase need not precede a statement that is the converse of a previous statement, unlike, say, "On the contrary,...". See, e.g., http://dictionary.cambridge.org/grammar/british-grammar/comparing-and-contrasting/contrasts

Page 4, lines 11-13: In this sentence, we have replaced the word "trends" with "decadal trends" to clarify that the statement refers specifically to the results cited in paragraph 2 on page 3. The results cited earlier in this paragraph are relevant to the results we derive later in the paper for Fraser-Hope, and this connection is made explicit at the beginning of Sec. 3.2.

Page 6, line 4: The abbreviation is now defined.

Page 6, lines 8-10: The MSS locations are not uniformly distributed across the FRB. We have added a short description to the sentence.

Page 6, line 12: SST has now been defined where it was first used, near the beginning of the paragraph.

Page 6, line 18: Added the year to this reference.

Section 2.2: We have included a few sentences describing the snow and soil routines, as requested by

the Reviewer, along with the appropriate references to the primary literature.

Page 7, line 8: We have clarified the reason why the frozen soil parameterization was not used.

Page 7, lines 10-17: In the second para on p. 7 it is now stated that the VIC simulation we analyzed was produced by colleagues for another purpose. For this reason, and since the calibration and validation procedures are thoroughly described in the papers referred to in the second para on p. 8, we feel that adding a description of the calibration and validation procedures here would be superfluous and detract from the flow of the paper.

Page 7, line 14: Islam and Dery (2017) used the Nash-Sutcliffe efficiency and temporal correlation coefficients as criteria for goodness-of-fit. However, we feel that it is unnecessary and possibly distracting to provide this level of detail in the text.

Page 11, lines 17-20: This CRI description was moved to Methods Sec. 2.3.

Page 13, lines 27-33: The few VIC grid cells with unrealistic SWE occur at or near high elevation locations in the FRB where glaciers are present in the real system. Thus, the deficiency lies not in the snow parameterization, but rather in the lack of a dynamic land-ice module that would permit the export of accumulated ice in one grid cell to its neighbours.

Page 14, lines 1-5: This paragraph was moved to Methods Sec. 2.2.

Page 15, lines 8-9: Yes, the results in Table 5 do suggest that large APFs in average SWE years are characterized by large dT/dt (1982, 1958, 2002). We have added a sentence to the first paragraph of Sec. 3.5.4 to emphasize this.

Page 15, lines 22 and 23: p-values inserted, thank you.

Page 18, line 6: We appreciate the Reviewer's suggestion, but beg to differ. We view the antecedent results in the paper as far more than a case study. Rather, they are a statistical summary of the APF-climate relationship in the FRB and its subbasins, which may apply to other nival watersheds as well. Consequently, we have left the title and organization of Sec. 4 unchanged.

Table 2: We agree, and replaced "Julian day" by "Calendar day"

Table 3: Changed "APDF" to "APF" in header

Figure 1: A scale bar was added to this figure, and the colour scheme changed to a more conventional green-to-brown for elevation.

Figures 2, 3, 5, 8 and 9: Where possible, we have enlarged the size of all fonts to improve legibility.

*Response to Reviewer 3 comments:*

1) Paper structure: In response to this criticism, and acknowledging similar comments from Reviewer 2, we have reorganized the content in the revised manuscript. Specifically, the descriptions of methods in the Results were moved to Section 2: Data and Methods.

2) We thank the referee for mentioning this additional data set for SWE. We agree that, were the focus of the paper on the predictability of APFs in an operational sense, i.e. flood forecasting, then incorporating these additional data would be a priority. However, as is now more clearly stated at several points in the paper, the intent of our work is rather different. Namely, we aim to discover and quantify relationships between the interannual variability of APF, large-scale climate indices, and various basin-averaged variables within the FRB. We feel that the data that we have used are adequate for this purpose; the MSS SWE data already demonstrate a strong connection between the annual max(SWE) and APF, and thus uncertainty in annual max(SWE) does not seem to be a significant factor. At the request of the Editor, we investigated the suitability of the snow pillow network records for incorporation in our study (also see below). Unfortunately, we found that the combined record of coincident data over the entire FRB spanned only 11 consecutive years, making it too short to be of use for the regression analysis. A sentence has been added to Sec. 2.1 of the revised manuscript to specifically address this point.

3) The selection of subbasins was made with three objectives in mind. The first was that all subbasins together capture a large fraction of the resultant flow at Fraser-Hope. As mentioned in the Abstract and in the first sentence of Sec. 3.6, the chosen subbasins collect nearly 70% of the observed mean annual streamflow at Fraser Hope (note that a reference to Kang et al. 2016, which provides the 70% estimate, has been added to the sentence). The second objective was to select basins that are sufficiently varied that they represent a range of elevations and, if possible, streamflow regimes. While the three larger subbasins are located in the eastern part of the FRB and represent nival environments, the Chilko basin lies on the westward edge of the FRB and integrates a significant amount of rainfall falling on the east-facing side of the Coast Mountains, making it a hybrid (nival-pluvial) environment. It was important to include this subbasin in order to probe the sensitivity of outlet streamflow to rainfall (although no such influence was detected on the APF in the Chilko). The third consideration was to avoid regulated basins, of which the Nechako-Nautley is one. In summary, although some medium-sized subbasins with sufficiently long data records were omitted from the study, we think it is unlikely that any important qualitative features of the climate-APF relationship were missed as a consequence.

4) We thank the referee for raising this thought-provoking point. We did not attempt to quantify the contribution of each subbasin to the overall APF at Fraser-Hope, nor did we calculate the lag between the respective peak flow days. We agree that information on the timing and magnitude of upstream flows would likely serve as effective predictors of APF at Fraser-Hope. However, as mentioned in point 2) above, the focus of our study is primarily on the climate-APF relationship on the interannual time scale, rather than short term prediction or monitoring of the upstream flow. We take this perspective not only to clarify the influence of historical climate variables on APF, but also to provide guidance for further study of the influence of changing climate on APF.

*Regarding the specific comments raised by Reviewer 3:*

1) We have added a sentence to the first paragraph of the Introduction pointing out the significance of the FRB for the salmon fishery in BC.

2) We have endeavoured to use abbreviations consistently after their first introduction in the manuscript.

3) We have removed this information from the Introduction, since it is already provided at the beginning of Section 2.

4) We agree with the referee and have removed the Dery et al. (2012) reference from this sentence, which refers mainly to the more recent paper. The former reference still appears in the next sentence.

5) Change made.

6) Noted. We request the editor's instruction on date formats for HESS.

7) We have corrected this and hopefully all other instances of missed unit superscripts in the revised ms.

8) Noted and changed, thank you.

9) Year added to this reference.

10) We have defined these abbreviations in the text.

11) The sentence concerning VIC elevation bands has been redrafted as follows: "The VIC implementation used in this study incorporates five elevation bands corresponding to 200 m vertical resolution, with the number of elevation bands in any one 1/16° grid cell depending on the topography within that cell."

12) We have added the suggested references to our initial discussion of teleconnections on p. 3.

13) Done.

14) As stated in the caption to Table 3, the trends in variables were calculated using Sen's median slope estimator, which is more robust to outliers than the standard least-squares method. This correction has been inserted at the beginning of Sec. 3.1.

15) We thank the referee for pointing out that most sites in the Upper Fraser studied in their previous paper have annual maximum SWE dates later than April 1. For our study, we only had access to the April 1 values from the MSS data, having obtained these data from a secondary source (Dr. R. Najafi). Since this was not made clear in the submitted MS, we have altered the data description in Sec. 2.1 to reflect this. More substantively, although using April 1 values might underestimate the true annual max(SWE), we do not believe this would bias the regression of max(SWE) versus APF, since years with larger/smaller than normal April 1 SWE should also feature larger/smaller than normal true max(SWE). Since it is the sign and relative magnitude of the annual max(SWE) anomaly, rather than its absolute value, that enters the regression analysis, we doubt that this would significantly alter the strong relationship found between observed SWE and APF discussed in Sec. 3.3.1.

16) Yes, change made.

17) Units have been inserted in Eqs. (1)-(3) (addressing point 19) also).

18) p-values inserted, thank you.

20)-25), 27) These changes were made.

26) The DOI has been added to this entry.

28) Our justification for the choice of subbasins was provided in the response to point 15) above. The period of record for each subbasin as listed in Table 1 reflects data availability from the WSC at the time of the analysis. However, note that observed discharge was only analyzed for the Fraser-Hope station, and not for the subbasin outlets.

29) The surface routing scheme in the VIC model does not account for any regulation of flows occurring in the real FRB. Consequently, the calibration procedure uses naturalized flow estimates in subbasins such as the Nechako where such diversions occur. For this reason, we can be confident that the difference between the mean hydrographs displayed in Fig. 5b are not attributable to this effect. Indeed, the difference is so small compared to the large interannual variability in both observed and VIC-simulated hydrographs that we are reluctant to conclude that the hydrographs are, in fact, significantly different.

*Response to Editor comments:*

We thank the Editor for his comments on our original Response to Reviewers, and for his additional suggested changes. We have addressed the questions of how our results apply to other cold regions dominated by snow and what these might mean for the future under continuing climate change in the revised Discussion and Conclusions section. Specifically, we discuss under what circumstances the derived MLR relations might be used to say something about projected future APFs.

We looked into the snow pillow measurements with a view toward integrating them in the study, but found that the records were, on the whole, too short to be of use in the regression analysis. We inserted a sentence in Sec. 2.1 addressing this specifically, i.e.:

"Another source of snow cover data, from automated snow pillows at high elevation sites, was also examined. However, as only a few of these sites have records longer than 20 years (within the 1956-2006 period of PCIC-OBS), we decided not to include them in the analysis."

In light of the high bias in SWE that is likely present in both MSS and snow pillow measurements (now mentioned at the end of Sec. 3.4), what is really needed for future studies in this area are snow amount measurements at low and medium elevation. Alas, we are not aware of such a data set for the FRB.

Since the original Response was submitted, we have made several further improvements to the paper, namely: 1) numerous clarifications and additional reorganization of the text in response to the Reviewers' suggestions; 2) improved versions of Fig. 2c and d, to better distinguish the streamflow in El Nino vs. La Nina years; 3) an improved colour scheme for the points in Fig. 4; 4) improved visibility of text in Fig. 8, and the addition of correlation coefficients between co-predictors.

Finally, we note one other change that was made to Figs. 2c and d in the revised manuscript. The quantile-quantile plots presented in the submitted manuscript did not clearly convey the number of APFs occurring in the different states: i.e., El Nino/La Nina and positive/negative PDO. In the revised ms, we rectified this by plotting the same data as a function of large-scale climate index state and percentile. The resulting figures are, in our opinion, more effective at conveying the difference in APF between different index states. The permutation test used to assess statistical significance is unchanged from that used in the original analysis.

We hope that these changes are considered sufficient for the acceptance of the revised manuscript by HESS.

[revised manuscript text omitted]

---

## Editor Decision (ED1)

Re-Review of "Examining controls on peak annual streamflow and floods in the Fraser River Basin of British Columbia"

Authors: Charles L. Curry and Francis W. Zwiers

Submitted in revised form to *Hydrology and Earth System Sciences*

Manuscript Number: HESS-2017-531

Summary: This revised manuscript remains a highly interesting and suitable article for possible publication in *Hydrology and Earth System Sciences*. The structure of the paper has been improved, the text is clear, and the figures depict key points discussed in the paper. Nevertheless, there remain some issues in regards to the inclusion of methods in the results section that need to be addressed before final publication of the article. This report therefore provides guidance for a few additional minor revisions that the authors should consider in preparing the final version of their manuscript.

The authors should note that it was difficult to follow the responses given that the three reviewers' original comments were not provided in the document. For future publications, the authors need to provide both the original comments and their responses in similar documents.

General Comments:

1) While the study focuses on the main stem Fraser River at Hope, BC that has an extended streamflow record, additional analyses are performed for four of its principal sub-watersheds (upper Fraser, Quesnel, Thompson and Chilko Rivers; see Table 1). These capture ~70% of the annual streamflow observed on the Fraser River's main stem at Hope, a statement reported in the abstract (p. 1, lines 20-21). Yet, no clear justification of the selection of these four additional sites is provided in Section 2.1, while other major tributaries to the Fraser are excluded in the present study? As stated in my previous report, the Nechako (Stuart/Nautley), Blackwater (West Road) and Chilcotin Rivers all form important tributaries to the Fraser River with generally complete observational records from the early 1950s onward (early 1970s for the Chilcotin River) and could provide further regional insights on the contributions of these systems to the APFs observed on the Fraser River at Hope.

2) Further to this, observed streamflow data for the Quesnel, Thompson and Chilko Rivers could easily be updated to 2014 to match the period of record for the Fraser River at Hope and at Shelley, BC.

3) Unfortunately, issues remain with the structure of the revised paper. Specifically, some of the methods used in the analyses are provided in the Results section, or are missing entirely from the Data/Methods section. Section 3.1 presents results of trend analyses on the input variables used in the VIC simulations, but nowhere in Section 2 are the methods for trend analyses discussed. Are trends inferred from linear regressions, the Mann-Kendall test, or another approach?

Section 3.3.3 focused on daily rainfall describes a method to deseasonalize the data, which also belongs in the Data and Methods section. Similarly, Section 3.4 that compare observed versus simulated streamflow and SWE is not well described previously in the Data and Methods section (e.g. the permutation test). Thus the authors need to revisit the entire paper and possibly undertake further restructuring to ensure methodological approaches are not introduced the Results section.

4) Some of the references are not in the appropriate journal format and/or lack important information such as range of pages/article numbers. In addition, some of the article titles are provided all in upper case letters, which is not required for HESS. The list of references must be carefully reviewed to ensure it matches the content and format requirements of the journal. Some specific issues are highlighted below.

5) Errors of up to $\pm12\%$ arise in measurements of streamflow during high flow conditions (e.g., Shiklomanov et al., 2006). How would such uncertainty affect the comparison between annual peak flows (APFs) observed in the FRB versus that simulated by the VIC model? At the very least, the authors should acknowledge that there are also possible errors in the observational record that could influence direct comparisons with simulated streamflow data.

Specific Comments:

1) P. 1, line 14 and elsewhere throughout the paper: The journal may request the format for dates be changed to "1 April".
2) P. 2, line 14: Delete "in order".
3) P. 2, line 16: Consider another term than the colloquial "home".
4) P. 3, lines 2-3: Please note that Padilla et al. (2015) provide the climatological month of peak flows at 141 gauging stations across the FRB that corroborates this statement.
5) P. 4, line 24: Replace "8" with "eight".
6) P. 4, line 26: Insert the years of publication for these references.
7) P. 5, line 20: Replace "8" with "eight".
8) P. 5, line 21: Replace "Center" with "Centre".
9) P. 6, line 7: Use upper case letter in naming the Variable Infiltration Capacity (VIC) model.
10) P. 6, line 31: Delete "In order".
11) P. 8, line 4: Equations may need to be numbered in the paper.
12) P. 12, line 7: Rephrase this sentence so that it begins "Three of the five…" and then insert in parentheses "(Fig. 4) at the end of the sentence.
13) P. 12, line 8: Replace "2" with "two".
14) P. 13, line 32: Replace the colloquial term "job".
15) P. 14, line 8: Insert a space between values and units (i.e. "2200 m").
16) P. 15, line 4: Spearman's rho is now in bold lettering while it was not in previous uses.

17) P. 15, line 7: Consider rephrasing this sentence given the repetition of words here ("found" and "find").
18) P. 15, line 23: Change to "three weeks".
19) P. 17, lines 27-28: Is the different response of the Chilko owing to its glacier melt dominated regime or its location on the eastern flanks and in the rain shadow of the Coast Mountains?
20) P. 18, line 3: Consider deleting "Moving now to the MLR analysis".
21) P. 18, line 8: Colloquial language again with "are at the heart".
22) P. 18, line 26: Should the delta term be deleted in "$\Delta dT/dt$"?
23) P. 18, line 31: Replace "is still" with "remains".
24) P. 20, line 26: "large" is used twice here, consider using "heavy snowpack" instead.
25) P. 21, line 1: The degree symbol is missing with the units of Celsius.
26) P. 21, line 8: Insert "to" before "its smoother".
27) P. 21, line 29: I doubt the area near Chilliwack, BC was "densely populated" 1894.
28) P. 22, line 6: Use superscripts only for units (i.e. "$m^3 \ s^{-1}$").
29) P. 22, line 32: Change to "seven of the top ten".
30) P. 25, line 21: Provide the article number for this reference.
31) P. 25, line 25: Has this paper now been published in the regular section of HESS?
32) P. 28, line 11: Insert the article number for this reference.
33) P. 28, line 14: This should be "Milly, P. C. D."
34) P. 28, line 26: Add the year of publication for this reference.
35) P. 29, line 10: The range of pages for this article is 588-592.
36) P. 32, Table 1: Why are the Upper Fraser, Quesnel, Thompson and Chilko Rivers selected as subbasins for this study? Why does the period of record end in 2013 for three of these rivers?
37) P. 33, Table 3: The methods used for trend analysis must be described in Section 2 with other methods.

References:

Padilla, A., Rasouli, K., and Déry, S. J.: Impacts of variability and trends in monthly runoff and water temperature on salmon migration in the Fraser River Basin, Canada, Hydrological Sciences Journal, 60, 523-533, doi:10.1080/02626667.2014.892602, 2015.

Shiklomanov, A. I., Yakovleva, T. J., Lammers, R. B., Karasev, I. P., Vörösmarty, C. J., and Linder, E.: Cold region river discharge uncertainty – estimates from large Russian rivers, J. Hydrol., 326, 231–256, 2006.

---

## Author Response (AR2)

***Note to Editor:***

Please see below for requested editorial advice in blue. Since the revised manuscript was submitted, we have made several additional editorial improvements to the paper, which appear in red font in the resubmitted manuscript. We hope that these responses and related changes are considered sufficient for the acceptance of the revised manuscript by HESS.

**Authors' response to Reviewer comments (second round)**

We thank the Reviewers and Editor for their additional comments on the revised manuscript. Our point-by-point response to each comment is given below, organized by reviewer. Revisions to the text of the manuscript based on the Reviewer comments are highlighted in red.

*Response to Anonymous Reviewer 1:*

1) (no response required)

2) *I'm still having trouble fully appreciating Figure 4; the third paragraph in section 3.3.3 doesn't help that much. Please explain what each point in the scatterplot represents – statistics for subsets of years associated with El Nino state, I assume? How many sample pairs go into these correlation calculations? The number must be small, making significance difficult to demonstrate.*

Response: Each point in the scatterplot represents the maximum value of the cross-correlation $\hat{\rho}_{XY}$ (as defined in Eq. [3]), calculated for a particular calendar year $i$ from 1955-2004. At each positive lag ($\tau$, in days), $\hat{\rho}_{XY,i}$ is calculated over $(365-\tau)$ pairs taken from the two daily time series for the $i^{th}$ calendar year. As explained after Eq. (3) in the text, only years for which $\max(\hat{\rho}_{XY}) > 2/\sqrt{365}$ at positive lag $\tau$ are shown (correlations in other years are deemed not significant). Finally, the corresponding ENSO state for that year is found, which determines the point colour. As now explained in Sec. 3.3.3, "For each point plotted in Fig. 4, the corresponding ENSO state for that year, taken from historical data, is indicated."

3) *p. 23, lines 11-12. Section 3 mentioned twice.*

Response: We have changed the last phrase of the sentence from "… in the regression analysis of Section 3" to "… in the FRB."

4) *p. 23, line 23 ("… following conditions were to hold."). Please clarify: both conditions, or either condition?*

Response: We have altered the phrase to read: "…if both of the following conditions were to hold."

*Response to Reviewer 2 (Michal Jenicek):* No response required.

*Response to Reviewer 3 (Stephen Déry):*

 *General comments:*

1) *"… no clear justification of the selection of these four additional sites is provided in Section 2.1, while other major tributaries to the Fraser are excluded in the present study? As stated in my previous report, the Nechako (Stuart/Nautley), Blackwater (West Road) and Chilcotin Rivers all form important tributaries to the Fraser River with generally complete observational records from the early 1950s onward (early 1970s for the Chilcotin River) and could provide further regional insights on the contributions of these systems to the APFs observed on the Fraser River at Hope."*

Response: Although we would respond similarly to this comment as in our initial response, we agree with the Reviewer that this justification should be provided in the text. We have therefore inserted the following in the second paragraph of Section 2.1:
"Daily streamflow data were obtained from the Water Survey of Canada (WSC) Hydrometric Database (HYDAT; Water Survey of Canada, 2016) for five hydrometric stations located within the FRB, as summarized in Table 1. The main outlet at Hope, which integrates the flows from all upstream locations, receives the most attention in the paper but four subbasin outlets are also considered. Three of these were selected on the basis of their leading contributions to the observed mean annual discharge at Fraser-Hope: Upper Fraser (29%), Thompson-Nicola (28%), and Quesnel (9%) (Kang et al., 2016). These subbasins are located in the eastern FRB, cover 45% of the total area, and represent nival environments. The smaller Chilko basin, by contrast, lies on the southwestern edge of the FRB and intercepts a significant amount of rain falling on the east-facing side of the Coast Mountains, making it a hybrid (nival-pluvial) catchment. Hence, this subbasin was included in an attempt to probe the sensitivity of streamflow to rainfall. Manual snow survey (MSS)…"

We settled on four subbasins as a representative, yet tractable, sample for additional analysis. As stated in the revised text, Chilko was selected as the fourth subbasin as an example of a hybrid (nival/pluvial) flow regime. Northern and Interior subbasins such as the Stuart, Nautley, and Blackwater (West Road) are, like the three eastern subbasins, snow-dominated. As mentioned in our earlier Response, the Nechako basin was excluded to avoid possible issues associated with regulation. The Chilcotin, which adjoins the Chilko, might have been a good alternative to the latter. However, as its outlet lies significantly to the east of the Chilko, we suspect it receives slightly less cold season rainfall. In addition, we should mention that the existence of a long measurement record is not a factor except for the Hope gauge station, because *the subbasin analysis in our paper is carried out for VIC-simulated streamflow only*. Thus, we reiterate that although some major tributaries of the Fraser were omitted from the study, we think it unlikely that any important qualitative features of the climate-APF relationship were missed as a consequence.

2) *Further to this, observed streamflow data for the Quesnel, Thompson and Chilko Rivers could easily be updated to 2014 to match the period of record for the Fraser River at Hope and at Shelley, BC.*

Response: As stated in our original response, the period of record for each subbasin as listed in Table 1 reflects data availability from the WSC at the time of the analysis. In any case, the point is effectively moot since observed discharge was only analyzed for the Fraser-Hope station, and not for the subbasin outlets, as mentioned above. The VIC-simulated, routed streamflow at the location of the subbasin outlets, on the other hand, was available only from 1955-2004. The date ranges in Table 1 are provided solely for informational purposes, in case others might be interested in follow-up work. (Note that in the interval since our analysis was conducted, WSC has updated the records at Spences Bridge and Hope to 2015 and 2016, respectively).

3) *"… some of the methods used in the analyses are provided in the Results section, or are missing entirely from the Data/Methods section. Section 3.1 presents results of trend analyses on the input variables used in the VIC simulations, but nowhere in Section 2 are the methods for trend analyses discussed. Are trends inferred from linear regressions, the Mann-Kendall test, or another approach? Section 3.3.3 focused on daily rainfall describes a method to deseasonalize the data, which also belongs in the Data and Methods section. Similarly, Section 3.4 that compare observed versus simulated streamflow and SWE is not well described previously in the Data and Methods section (e.g. the permutation test). Thus the authors need to revisit the entire paper and possibly undertake further restructuring to ensure methodological approaches are not introduced the Results section."*

Response: The method for deriving temporal trends, the Theil-Sen median slope estimator, is now described in Sec. 2.3, as is the permutation (resampling) test subsequently applied in Secs. 3.2 and 3.4. In Sec. 3.4, the permutation test is not used for the SWE comparison, which is primarily visual. Regarding the rainfall-APF analysis of Sec. 3.3.3, we first clarify that the cross-correlation function defined there (now labelled as Eq. [3]) is not a method to deseasonalize the data; rather, it is calculated from such data. Second, methods that are applied in a single instance and that are also in fairly common use are, we feel, better described immediately prior to their application. That is, readability is enhanced when such a method is mentioned and then directly applied to obtain results (as opposed to situations where the description of a method would require a lengthy digression). We maintain that this is the case in Sec. 3.3.3 and so have elected not to shift this material back to Sec. 2.3.

4) *Some of the references are not in the appropriate journal format and/or lack important information such as range of pages/article numbers. In addition, some of the article titles are provided all in upper case letters, which is not required for HESS. The list of references must be carefully reviewed to ensure it matches the content and format requirements of the journal. Some specific issues are highlighted below.*

Response: We thank the Reviewer for pointing out these bibliographic issues. We have reviewed the references and corrected and provided additional information where necessary.

5) *Errors of up to ±12% arise in measurements of streamflow during high flow conditions (e.g., Shiklomanov et al., 2006). How would such uncertainty affect the comparison between annual peak flows (APFs) observed in the FRB versus that simulated by the VIC model? At the very least, the authors should acknowledge that there are also possible errors in the observational record that could influence direct comparisons with simulated streamflow data.*

Response: We thank the Reviewer for this information and agree that measurements of individual APFs have associated errors. It seems reasonable to conclude that such errors are not systematic, however, and sum to a small value over the 50-year analysis period considered here. Recall that our evaluation of VIC in Sec. 3.4 considers both the multi-year mean and interannual variance of the observed and modelled distributions of APF. Over the analysis period, the standard deviation of observed APF at Hope is 1581 $m^3s^{-1}$, or 18% of the multi-year mean observed APF. This figure is greater than the difference between modelled and observed multi-year means (8%, with VIC larger), and also, we suspect, considerably larger than the summed errors in individual APF observations (although these could be important in a year-by-year comparison). The fact that the VIC calibration uses a subset of the same observations to fix specifiable parameters in the model also likely accounts for this overall level of agreement (Shrestha et al., 2012). Thus, we have summarized this state of affairs as follows in the first paragraph of Sec. 3.4: "… given that the interannual coefficient of variation in observed APF (CV = $\sigma$/mean[APF]) is 18%, that the VIC-simulated CV = 20%, and their degree of overlap, we conclude that the hydrographs are not significantly different. "

*Specific comments raised by Reviewer 3:*

1) *p.1, line* 14: We request the Editor's instruction regarding date conventions for HESS.

2) *p.2, line 14*: Deleted "in order."

3) *p.2, line 16*: We feel that "home" is not used in a colloquial sense in this sentence, so have left it as is.

4) *p.3, lines 2-3*: We added a reference to Padilla et al. (2015) in the text and bibliography.

5) *p.4, line 24*: "8" changed to "eight."

6) *p.4, line 26*: We inserted the years of these publications.
7) *p.5, line 20*: "8" changed to "eight."

8) *p.5, line 21:* "Center" changed to "Centre."

9) *p.6, line 7*: Changed expanded VIC to capital letters.

10) *p.6, line 31:* We appreciate the Reviewer's suggestion, but have elected to keep "In order to" at the beginning of this sentence for stylistic reasons.

11) *p.8, line 4*: As suggested, we have numbered this equation and renumbered all subsequent equations and references thereto.

12) *p.12, line 7*: We adopted this stylistic suggestion.

13) *p.12, line 8*: "2" changed to "two."

14) *p.13, line 32*: Changed this phrase to "… VIC simulates APF reasonably well compared to observations."

15) *p.14, line 8*: A space was inserted, i.e., "2200 m."

16) *p.15, line 4*: Boldface font removed from Spearman $\rho$.

17) *p.15, line 7:* Changed "found" to "noted."

18) *p.15, line 23*: Changed "3" to "three."

19) *p.17, lines 27-28*: We have expanded upon this result as follows: "The weak dependence of streamflow on the PDO, SOI and Pacific-North American indices in this and other catchments in the western FRB was also noted by Thorne and Woo (2011), who attributed this insensitivity to the low magnitude of discharge in these tributaries. These authors speculated that the main trunk of the Fraser, by contrast, integrates the influence of regional climate forcings on upstream catchments, bolstering the teleconnections."

20) *p.18, line 3*: We believe that the connecting phrase "Moving now to the MLR analysis," enhances the readability of the text, so have elected to retain it.

21) *p.18, line 8*: We replaced "at the heart of" by "integral to."

22) *p.18, line 26*: No, $\Delta dT/dt$ is what was intended, as it refers to the relative anomaly of *dT/dt*. However, we have improved the notation here and in the preceding paragraph. First, we augmented the definition of the relative anomaly at line 20 as follows: $\Delta X/\overline{X} = (X_i - \overline{X})/\overline{X}$. Second, for clarity we denote $\Delta dT/dt$ instead as $\Delta(dT/dt)$.

23) *p.18, line 31*: Replaced "is still" with "remains."

24) *p.20, line 26*: Replaced "an abnormally large snowpack" with "a heavy snowpack."

25) *p.21, line 1*: Inserted degree symbol.

26) *p.21, line 8*: Inserted "to."

27) *p.21, line 29*: Replaced "densely populated" with "well populated."

28) *p.22, line 6*: Replaced "m$^3$/s" with "m$^3$ s$^{-1}$"

29) *p.22, line 32*: Replaced "7" with "seven."

30) *p.25, line 21*: Article number was added.

31) *p.25, line 25*: Presuming the Reviewer meant "line 22", no, this article has not yet appeared in HESS.

32) *p.28, line 11*: There does not seem to be an article no. assigned to McCabe et al. (2007).

33) *p.28, line 14*: Initial added to this entry.

34) *p.28, line 26*: We have removed this reference from the bibliography, as it was incorrect. Also, a brief literature search shows that similar data sets are referred to by URL in the main text. Hence, we have inserted a URL in the text on p.7, line 19. The Editor can let us know if this needs revision.

35) *p.29, line 6*: Inserted missing page number.

36) *p.32, Table 1*: These questions are addressed in General Comment 1) above.

[revised manuscript text omitted]

---

## Author Response (AR3)

**Authors' response to Editor comments**

We thank the editor for his latest suggestions to improve our manuscript. A point-by-point response is given below. A few minor revisions to the text and figures have also been made in the meantime. All changes since the last revision are shown in the manuscript in blue font.

*1) Reviewer #1 has expressed concerns about Figure 4 and the CRI and how clear the description and interpretation of these are. The responses provided in the author comments after both rounds of review provide a clear explanation, particularly the more recent response explaining what each point in the plot represents. It would be beneficial to include some of this within the final paper to help the readers better understand the concepts.*

**Response:** Agreed. We have added this explanatory text at line 24-25 on p.12 of the resubmitted MS.

*2) Regarding date formats for HESS, at
https://www.hydrology-and-earth-system-sciences.net/for_authors/manuscript_preparation.html->
Mathematical requirements ->
Date and time there is the following information:
25 July 2007 (dd month yyyy), 15:17:02 (hh:mm:ss). Often it is necessary to specify the time if referring to local time or universal time coordinated. This can be done by adding "LT" or "UTC", respectively.*

*Please change the dates throughout the manuscript to be consistent with this.*

**Response:** Thank you, we have changed the dates throughout the MS accordingly.

*3) Regarding insertion of the URL (https://www.pacificclimate.org/data/station-hydrologic-model-output) on page 7, line 25, this is acceptable. However, note that on that webpage it states: "When referring to the Station Hydrologic Model Output data retrieved from the website or found otherwise, the source must be clearly stated: Pacific Climate Impacts Consortium, University of Victoria, (Jan. 2014). Station Hydrologic Model Output. Downloaded from on ."*

**Response:** We have changed in inline reference to "(Pacific Climate Impacts Consortium, 2014)" and placed the URL and date of access in the Bibliography.

*4) Page 8, line 15: Please correct the grammar in the phrase "Trends were detected in certain of the time series analysed in this work..."*

**Response:** This has been simplified to: "Trends were detected in certain time series, as reviewed..."

*5) Page 8, lines 18-20: Should a consistent tense be used here? i.e. "If a significant relationship was found... then the coefficient is taken..." You may use your discretion.*

**Response:** Changed "is taken" to "was taken" in this sentence.

*6) Pages 9 and 10: Section 3.1, 3.2, and 3.3 all start with "We begin", which is repetitive. You may want to revise this, but it is up to your discretion.*

**Response:** We have introduced some word variety at the beginning of Secs. 3.2 and 3.3.

*7) Page 10, line 22: Should it not say "These plots" rather than "This plot"?*

**Response:** Agreed, change made.

*8) Pages 20 and 21, case studies: It would be helpful to note for each of the three years the rank of the observed APF, in addition to that of the simulated APF. For instance, 1972 had the highest observed APF, and 1999 had the 5th highest APF (Fig. 5a). This helps remind the reader of the difference between observations and model results, especially given that in Section 4.4 you are referring to observed/estimated APF for the two highest flow years.*

**Response:** This information has been added to the text of Sections 4.1 and 4.3 (it was already in Sec. 4.2).

*9) Page 43, Figure 8: the figure would be easier to interpret if each graph were labeled by basin or by alphabetic notation, as with the other figures. Also, the red to green colour scale may be difficult to interpret for some readers with this colour blindness. You may use your discretion.*

**Response:** Each panel is in fact labelled by subbasin abbreviation at upper right. We have improved the visibility of this notation by using boldface font. The colour scale has been changed to blue-brown to address the colour-blindness issue, and the revised figure added to the MS.

[revised manuscript text omitted]